# UQ-Guided Hyperparameter Optimization for Iterative Learners

**Jiesong Liu◇, Feng Zhang⋆, Jiawei Guan⋆, Xipeng Shen◇,**
◇Department of Computer Science, North Carolina State University
⋆School of Information, Renmin University of China
jliu93@ncsu.edu, guanjw@ruc.edu.cn, fengzhang@ruc.edu.cn, xshen5@ncsu.edu

## Abstract

Hyperparameter Optimization (HPO) plays a pivotal role in unleashing the potential of iterative machine learning models. This paper addresses a crucial aspect that has largely been overlooked in HPO: the impact of uncertainty in ML model training. The paper introduces the concept of *uncertainty-aware HPO* and presents a novel approach called the *UQ-guided scheme* for quantifying uncertainty. This scheme offers a principled and versatile method to empower HPO techniques in handling model uncertainty during their exploration of the candidate space. By constructing a probabilistic model and implementing probability-driven candidate selection and budget allocation, this approach enhances the quality of the resulting model hyperparameters. It achieves a notable performance improvement of over 50% in terms of accuracy regret and exploration time.

## 1 Introduction

Hyperparameter optimization (HPO) is essential for unleashing the power of iterative machine learning models [4, 18, 40]. Hyperparameters include traditional parameters like learning rates and more complex ones like neural architectures and data augmentation policies. For iterative learners (multi-fidelity and early stopping methods specifically), practitioners can obtain intermediate validation loss after each iteration and use them for model assessment; the main goal of HPO is to explore a vast candidate space to find candidates that lead to optimal model performance.

There are many designs in the literature to solve the HPO problem. Successive Halving (SH) [19], for example, terminates training of candidate configurations with poor performance early so as to save computing resources for more well-behaved candidates. Bayesian optimization [17, 34], another optimization method, uses a surrogate model to guide the selection of candidate configurations for assessment.

There is however a lack of systematic treatment of an important factor in HPO designs, the uncertainty inherent in the dynamics of the training process of iterative machine learning applications. Because of the uncertainty, a model with a candidate hyperparameter configuration (or candidate in short) performing poorly in an early stage of its training could turn out to be the best model after convergence. Such candidates are however likely to be stopped from proceeding further or be completely discarded by existing HPO methods in the early stages, because their selections of candidates are mostly based on the currently observed performance, for lack of a way to treat the uncertainty properly. In 100 experiments of Successive Halving, for instance, the actually best candidates were discarded in the first 8–22 iterations of training, causing 48% performance regrets in validation loss (details in Section 3.1 Figure 1 and Section 4 Figure 3).

This paper introduces model uncertainty into the design of HPO methods for iterative learners and establishes the concept of *uncertainty-aware HPO*. At the core of uncertainty-aware HPO is a novel uncertainty quantization (UQ) guided scheme, named *UQ-guided scheme*, which unifies the selection of candidates and the scheduling of training budget—two most important operations in HPO—into a single UQ-based formal decision process. The UQ-guided scheme builds on a probabilistic uncertainty-based model, designed to approximate the statistical effects of discarding

a set of candidates at the end of a step in HPO. It uses a lightweight method to efficiently quantify model uncertainty on the fly. It offers a principled, efficient way for HPO to treat model training uncertainty.

As a general scheme, the *UQ-guided scheme* can be integrated into a variety of HPO methods for iterative learners, especially DNNs. This paper demonstrates its usefulness and generality for DNNs by integrating it into four existing HPO methods. Experiments on two widely used HPO benchmarks, NAS-BENCH-201 [9] and LCBench [42], show that the enhanced methods produce models that have 21–55% regret reduction over the models from the original methods at the same exploration cost. And those enhanced methods need only 30–75% time to produce models with accuracy comparable to those by the original HPO methods. The paper further gives a theoretical analysis of the impact of the *UQ-guided scheme* for HPO.

## 2   Background and Related Work

Many studies are committed to solving the HPO problem for iterative learners efficiently [26, 20, 29, 39, 41]. Bayesian optimization, early stop-based mechanisms, and multi-fidelity optimizations are some important approaches.

**Bayesian Optimization (BO).** BO is a sequential design strategy used to optimize black-box functions [34, 17, 11]. In HPO scenarios, it can be used as a surrogate model to sample high-quality candidates.

**Early Stop Mechanisms.** Early stop-based approaches can be effective since they evaluate different candidates during training and make adaptive selections accordingly [2, 8, 35]. The early stopping mechanism, which stops the training of poorly-performed candidates early, has been widely employed in the HPO community including Successive Halving (SH) [19] and Hyperband (HB) [25]; BOHB [11] combines both BO and HB methods to take advantage of both the BO surrogate model and the early stopping mechanism.

**Multi-fidelity Optimizations.** Multi-fidelity evaluation focuses on using low-fidelity results trained with small resources to accelerate the evaluation of candidates [2, 3, 8, 22, 23, 38, 21, 35, 36, 14, 26]. Sub-sampling (SS) [15] is proposed mainly using multi-fidelity methods to collect high-quality data to select good configurations without early stopping.

**Model Uncertainty in HPO.** Various optimization methods in HPO scenarios focus on specific training metrics to assess candidate performance. However, these methods typically overlook the uncertainty in the candidate selection process. Machine learning models inherently have approximation uncertainties [5, 6, 10, 12, 24, 28, 31]. Some HPO designs sample the candidate space based on distributions on the effect of each hyperparameter dimension on the quality of the candidates, but without considering the uncertainty in the model training process. For example, one of the studies [33] separates candidates into "good" or "bad" groups in order to build the distributions. The separation is based on the same deterministic metrics as other HPO methods use, giving no consideration of the uncertainty in model trainings. The only work we find that considers uncertainty in training metrics [32] selects configurations for further training based on its assessment of the upperbound of those configurations. In each round, the configurations it chooses are those that, considering the best possible performance at the last iteration, show a smaller validation loss than the validation loss current best configuration shows. The selection treats model uncertainty preliminarily and does not use it to guide the allocation of training budget. We compare other related works in Appendix E.

## 3   Uncertainty Quantification (UQ)-Guided Hyperparameter Optimization

This section gives an exploration of model uncertainty, introduces *UQ-guided scheme* for incorporating UQ into the design of HPO, discusses examples of ways to use the UQ-guided scheme to enhance existing HPO methods, and theoretically analyzes its effects.

### 3.1   Uncertainty in Iterative Machine Learning

Uncertainty in iterative machine learning originates mainly from two factors: inherent noise in the data and the variability of model predictions due to restricted knowledge [16, 1]. Since data uncertainty is constant, it is the variability in model predictions, referred to as model uncertainty, that primarily influences decisions on HPO. To estimate the model uncertainty, we can incur small perturbations to the model, evaluate these model variants, and calculate the variance of the results as the approximation for the model uncertainty [16].

Figure 1 shows how the model uncertainty affects the quality of the returned candidate. In a given SH run, half of the candidates are eliminated at each checkpoint marked by a vertical red dashed line. The solid blue line represents the best validation loss up to the current point, while the orange dashed line signifies the true quality (in terms of validation loss after convergence) of the candidates SH retains at that specific juncture. From the figure, we see that in every round, SH discards the actually best candidates, causing a continuous increase of the regret. The reason is that the discarding decision of SH is solely based on the current validation loss, but model uncertainty, particularly pronounced in the early stages, obfuscating the true model potential.

### 3.2 Quantify Uncertainty and the Impact

We explain how we quantify model uncertainty, and how, based on it, at any point of time, estimate the performance and uncertainty of a candidate model when its training converges.

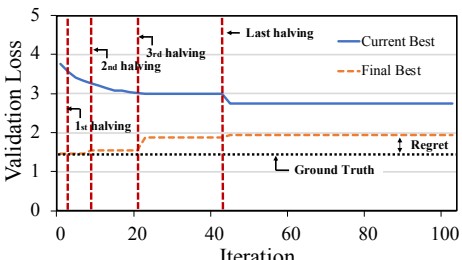

High efficiency is essential here as the UQ happens during the HPO process. We employ a lightweight approach to conduct the UQ efficiently on the fly, as explained next.

Let $\gamma_1, \gamma_2, \cdots, \gamma_K \in \Gamma$ be $K$ candidates drawn from the hyperparameter space $\Gamma$. Consider a supervised learning setting, where a machine learning model $M$ is trained on some training set $D_{Train} = \{(\mathbf{x}_1, y_1), (\mathbf{x}_2, y_2), \cdots, (\mathbf{x}_{n_{train}}, y_{n_{train}})\}$. Let $M_\gamma^t$ denote the model with hyperparameter $\gamma$ trained on $D_{Train}$ after $t$ epochs, and $M_\gamma^*$ the converged model. $M_\gamma^t(\mathbf{x})$ gives the prediction on a given input $\mathbf{x} \in \mathbb{R}^d$.

Figure 1: Demonstration of the negative impact from uncertainty on HPO; Successive Halving (SH) is used; the benchmark is NAS-BENCH-2.0 [9]. More detailed are in Section 4.1 and Appendix F. Due to its overlooking at model uncertainty, at each halving point, SH discards the actual best candidates, causing an increase in the regret.

We use $\ell(\cdot, \cdot)$ to denote the metric that evaluates the performance of a candidate model. Given a hyperparameter configuration $\gamma_c$, the validation loss of a model instance, $\ell(\mathbf{y}, M_{\gamma_c}^*(\mathbf{X}))$, can be affected by training data and other factors, and hence has some uncertainty. Let $\mathcal{N}(\mu_c*, \sigma_c^2*)$ represent the distribution of $\ell(\mathbf{y}, M_{\gamma_c}^*(\mathbf{X}))$, where, $\sigma_c^2*$ embodies the uncertainty.

Our objective here is a way that can, at any point in the HPO process, estimate $\sigma_c^{*2}$ and $\mu_c^*$ for a candidate model configured with $\gamma_c$. Our solution leverages the validation history in the HPO process, $\ell(\mathbf{y}, M_{\gamma_c}^1(\mathbf{X})), \cdots, \ell(\mathbf{y}, M_{\gamma_c}^t(\mathbf{X}))$, to construct an estimated loss curve, explained as follows.

**Decomposition of momentum and the underlying structure of the metric.** This part uses breakdowns to characterize the loss curve and introduces the objective function we want to minimize to estimate the curve parameters. The first component, referred to as momentum, models the decaying trend of the loss curve. The second component is the bias term for each candidate's loss curve; it models a latent effect underlying the hyperparameter space by allowing correlation among the candidates. The details are as follows.

Typically a candidate in HPO can be represented as a vector. We use a set $\mathcal{U}_r \subset \mathbb{R}^r$ to represent the candidates $\{\gamma_i, i \in [n]\}$; $r > 0$ is the vector length. For modeling the loss curve, we set $\mathbf{k}_t \in \mathbb{R}^L$ as a vector whose elements are functions of training epochs $t = 1, 2, \cdots$. (In our experiments, we set $\mathbf{k}_t = [t^{-1/2}, t^{-1}]$, for the general decreasing trend of loss curves as training epochs increase.) We model metric $\ell_t^{\mathbf{u}}$ of candidate $\mathbf{u} \in \mathcal{U}_r$ at time $t$ as the summation of the impact from three sources:

$$\ell_t^{\mathbf{u}} = \ell(\mathbf{y}, M_{\gamma_c}^t(\mathbf{X})) = \mathbf{k}_t^\top \boldsymbol{\eta}^{\mathbf{u}} + \mathbf{u}^\top \mathbf{Z} + \epsilon_t^{\mathbf{u}}, \qquad (3.1)$$

where $\boldsymbol{\eta}^{\mathbf{u}}$ and $\mathbf{Z}$ are the parameter vectors to determine. The three components in Equation 3.1 are (1) the momentum part ($\mathbf{k}_t^\top \boldsymbol{\eta}^{\mathbf{u}}$) that denotes the contribution from the trends in the loss curve of the training specific candidate model variant, (2) the contribution from the underlying model structure ($\mathbf{u}^\top \mathbf{Z}$), and (3) the noises by other elements $\epsilon_t^{\mathbf{u}}$, which is asssumed to follow a Gaussian distribution $\mathcal{N}(0, \sigma_t^2)$ independent of each other and of $\mathbf{Z}$.

For a candidate $\mathbf{u} \in \mathcal{U}_r$, let $\alpha^{\mathbf{u}} = \mathbf{u}^\top \mathbf{Z}$. The loss curve parameters $\{\alpha^{\mathbf{u}}, \boldsymbol{\eta}^{\mathbf{u}}\}$ can be determined by optimizing a weighted least squares objective

$$\mathcal{G} = \min_{\alpha^{\mathbf{u}}, \boldsymbol{\eta}^{\mathbf{u}}} \sum_{t=1}^{T} \sum_{j=1}^{F_k} w_{jt} (\ell_{jt}^{\mathbf{u}} - \alpha^{\mathbf{u}} - \mathbf{k}_t^{\top} \boldsymbol{\eta}^{\mathbf{u}})^2 \tag{3.2}$$

where $w_{jt} = \frac{1}{F_t \sigma_t^2}$. $F_t$ is the number of models trained with time $t$.

**Solving for the Momentum Mean and Variance.** This part analyzes the mean and variance of the loss curve we constructed and makes inferences to the converged loss based on the loss curve. The quantified uncertainty of the estimated converged loss is then used for model selection and budget resource allocation. The specifics are as follows.

For a candidate $i$ at iteration $T$, concatenating the validation losses across training epochs (indexed by $t$) will lead to a validation loss vector $\mathbf{v}$ of dimension $D = \sum_{t=1}^{T} F_t$. For each $d \in \{1, \cdots, D\}$, the $d^{th}$ element in $\mathbf{v}$ is an observation of loss at time $t_d$ that follows $\mathcal{N}(\mu_{i_{t_d}}, \sigma_{t_d}^2)$ with $t_d$ mapping $d$ to its corresponding epoch $t$.

For a given candidate $\mathbf{u}$ (for better readability, we omit the superscript $\mathbf{u}$ in the notations in the following discussion), the weighted least squares problem can be formulated as solving the equation $\mathbf{W}^{\frac{1}{2}} \mathbf{v} = \mathbf{W}^{\frac{1}{2}} \mathbf{A} \boldsymbol{\beta}$ for $\boldsymbol{\beta}$ with $\mathbf{W} \in \mathbb{R}^{D \times D}$ being a diagonal matrix of weights $\mathbf{W}_{dd} = \frac{1}{F_{t_d} \sigma_{t_d}^2}$. $\mathbf{A} = [\mathbf{1} \quad \mathbf{K}]$, $\mathbf{K} \in \mathbb{R}^{D \times (L+1)}$ with $\mathbf{K}[d, :] = \mathbf{k}_{t_d}^{\top}$, and $\boldsymbol{\beta}^{\top} = [\alpha \quad \boldsymbol{\eta}^{\top}]$. The empirical estimate of $\sigma_i^2$ is computed as the variance of the loss in the recent several epochs of $i$—those instances can be regarded as results of small perturbations to the model at epoch $i$.

Solving the weighted least square objective, we have the estimator as $\hat{\boldsymbol{\beta}} = (\mathbf{A}^{\top} \mathbf{W} \mathbf{A})^{-1} (\mathbf{A}^{\top} \mathbf{W} \mathbf{v})$. The covariance of the estimator is given by $\Sigma_{\hat{\boldsymbol{\beta}}} = (\mathbf{A}^{\top} \mathbf{W} \mathbf{A})^{-1}$.

Since the estimated curve is given by $\hat{v}(t) = [1 \quad \mathbf{k}_t^{\top}] \hat{\boldsymbol{\beta}}$, the variance of this estimation is given by

$$\hat{\sigma}^2(t) = [1 \quad \mathbf{k}_t^{\top}] \Sigma_{\hat{\boldsymbol{\beta}}} \begin{bmatrix} 1 \\ \mathbf{k}_t \end{bmatrix}.$$

With the formulas for $\hat{v}(t)$ and $\hat{\sigma}^2(t)$, we can then approximate the distribution of $\ell(\mathbf{y}, M_{\gamma_i}^*(\mathbf{X}))$ as $\mathcal{N}(\hat{v}(N), \hat{\sigma}^2(N))$ for a large $N$ value ($N = 200$ in our experiments).

**Algorithm.** Based on the analysis, we devise an iterative algorithm to compute the estimated loss and variance. Without the loss of generality, we make the following assumption.

**Assumption 1.** *There exist positive constants $\overline{u}$ such that for any $r$, $\max_{\mathbf{u} \in \mathcal{U}_r} \|\mathbf{u}\| \leq \overline{u}$, and the set of candidates $\mathcal{U}_r \subset \mathbb{R}^r$ has $r$ linearly independent elements $\mathbf{b}_1, \cdots, \mathbf{b}_r$.*

The algorithm goes as follows. At the beginning of HPO, it sets $\mathbf{Z}$ with a random vector. At the end of each epoch, it conducts the following two operations. First, it solves the weighted least squares objective 3.2 for each of the $r$ candidates (mentioned in Assumption 1) by following the formulas described earlier in this section, with the current $\mathbf{Z}$ value being used. Second, for $p = 1, 2, \cdots r$, it observes the metrics $X^{\mathbf{b}_p}(t) = \ell_t^{\mathbf{b}_p} - \mathbf{k}_t^{\top} \hat{\boldsymbol{\eta}}^{\mathbf{b}_p}$ and refines the ordinary least square estimate for $\mathbf{Z}$ as follows:

$$\widehat{\mathbf{Z}} = \left( \sum_{p=1}^{r} \mathbf{b}_p \mathbf{b}_p^{\top} \right)^{-1} \sum_{p=1}^{r} \mathbf{b}_p X^{\mathbf{b}_p}$$

At the end of an HPO round (e.g., at the halving time in SH), it performs the first step for every candidate model to compute the estimated distributions of their $\ell(\mathbf{y}, M_{\gamma_i}^*(\mathbf{X}))$, so that the HPO can use the estimates to select promising candidates to continue in the next round.

We next show how to use the approximated loss and uncertainty to compare two candidates:

**Definition 1** (UQ-guided comparison of candidates). *UQ-guided comparison of candidates compares two candidates based on the probability that the validation loss of the converged model $\gamma_{c_1}$ is lower than that of $\gamma_{c_2}$, represented as follows based on the approximation from the current validation losses and uncertainty of the two candidates:*

$$P = \Pr \left\{ \ell(\mathbf{y}, M_{\gamma_{c_1}}^*(\mathbf{X})) > \ell(\mathbf{y}, M_{\gamma_{c_2}}^*(\mathbf{X})) \Big| \ell(\mathbf{y}, M_{\gamma_{c_1}}^{t=1,2,\cdots}(\mathbf{X})), \hat{\sigma}_{c_1}, \ell(\mathbf{y}, M_{\gamma_{c_2}}^{t=1,2,\cdots}(\mathbf{X})), \hat{\sigma}_{c_2} \right\}. \tag{3.3}$$

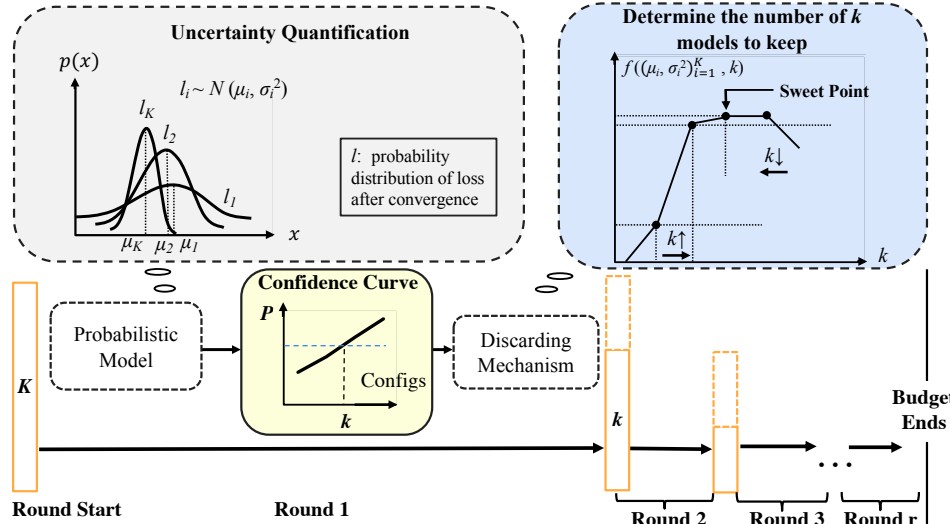

Figure 2: Illustration of using UQ-guided scheme to enhance Successive Halving. The goal is to select an optimal hyperparameter configuration from $K$ candidates. It involves multiple rounds. $R$ is the predefined budget resources (e.g., training epochs) for each round. For the first round, $K$ candidates each get trained for $\frac{R}{K}$ epochs. Based on the observed validation loss and the quantified uncertainty for each candidate, our method represents each candidate's converged loss with a probability distribution. From that, it constructs a *confidence curve*, capturing the probability that the best configuration is among the current top $k$ candidates for $1 \leq k \leq K$. From the curve, it then calculates $f((\mu_i, \sigma_i^2)_{i=1}^K, k)$, which captures the effects of keeping $k$ top candidates ($1 \leq k \leq K$) for the next round, by considering the tradeoff between the risks of discarding the best candidate and the training budget each top candidate can get. From that, it identifies the best $k$ value, discards the least promising $K - k$ candidates, and enters the next round. The process continues until the total budget is used up.

The main idea of Definition 1 is to use the current validation loss history and quantified uncertainty to approximate the converged validation loss, so that we compare two candidates – more precisely, we compute the probability that one candidate is better than the other – based on the probability distribution of their converged validation loss. For example, if the approximated loss and uncertainty of two candidates $\gamma_j$ and $\gamma_k$, at epoch $t$, are $(\mu_j, \sigma_j)$ and $(\mu_k, \sigma_k)$, using converged validation loss as the metric, we have $\Pr(\ell(\mathbf{y}, M_{\gamma_j}^t(\mathbf{X})) > \ell(\mathbf{y}, M_{\gamma_k}^t(\mathbf{X}))) = \Phi\left(\frac{\mu_k - \mu_j}{\sqrt{\sigma_j^2 + \sigma_k^2}}\right)$, where $\Phi$ denotes the cumulative distribution function (CDF) of the standard normal distribution.

We next present *UQ-guided scheme*, a principled way to use UQ to guide HPO.

### 3.3 UQ-Guided Scheme

Figure 2 illustrates how UQ-guided scheme works in HPO. For the purpose of clarity, we base our explanation of the scheme on HPO that uses early stop mechanisms, but will show in Section 3.5 that the scheme is general, applicable to other HPO methods for iterative learners as well.

The original HPO method, Successive Halving [19], evaluates and eliminates candidates over multiple rounds. At the end of each round, it drops those candidates regarded as unpromising. With our UQ-guided scheme, at the end of each round, the scheme derives a *confidence curve* from the current probabilistic model, and uses a *discarding mechanism* to drop candidates that are unlikely to perform well after convergence. In contrast to the original HPO that drops a fixed amount (or fraction) of candidates in each round, the UQ-guided scheme carefully calculates the number of candidates to drop in a round based on the probabilistic model such that the expected quality of the HPO outcomes can be maximized, as explained later.

The UQ-guided scheme respects the HPO budget—that is, the total amount of time usable by the HPO for identifying the best candidate. By default, it works around the given budget constraint: the budget for each round ($R$) equals the total budget divided by the number of rounds. We next discuss each step.

#### 3.3.1 Confidence Curve Derived from Uncertainty Quantification

The concept of *confidence curve* is central in UQ-guided HPO. Define $[n] = \{1, 2, \cdots, n\}$.

**Definition 2** (Confidence curve). *At epoch t, we evaluate each candidate's performance and sort them based on validation loss. A* confidence curve $\mathcal{C}$ *is a trajectory of a series of probabilities,* $\{P_k | k \in [n]\}$, *that depicts the probability that the optimal configuration (with the lowest loss after convergence) is among the first $k$ configurations. For $k \in [n]$, $P_k$ can be expressed as*

$$P_k = \Pr \left\{ \min(\ell(\mathbf{y}, M^*_{\gamma_1}(\mathbf{X})), \cdots, \ell(\mathbf{y}, M^*_{\gamma_k}(\mathbf{X}))) \leq \min(\ell(\mathbf{y}, M^*_{\gamma_{k+1}}(\mathbf{X})), \cdots, \ell(\mathbf{y}, M^*_{\gamma_n}(\mathbf{X}))) \right\}.$$

The *confidence curve* is derived based on joint probability distribution in the following way. Suppose there are $n$ candidates. At the end of a certain round, the probabilistic model returns $n$ pairs of $(\mu_1, \sigma_1), (\mu_2, \sigma_2), \cdots, (\mu_n, \sigma_n)$ as estimations for $\ell(\mathbf{y}, M^*_{\gamma_1}(\mathbf{X}))$, $\ell(\mathbf{y}, M^*_{\gamma_2}(\mathbf{X}))$, $\cdots$, $\ell(\mathbf{y}, M^*_{\gamma_n}(\mathbf{X}))$. For simplicity, assume that $\mu_1 < \mu_2 < \cdots < \mu_n$, and $\sigma_1 = \sigma_2 = \cdots = \sigma$.

Let $\Phi$ and $\phi$ be the CDF and PDF of the standard normal distribution. For $k = m$, we can calculate

$$P_m = \int_{-\infty}^{\infty} \frac{1}{\sigma} \sum_{i=1}^{m} \frac{\phi(\frac{y+\mu_i}{\sigma})}{\Phi(\frac{y+\mu_i}{\sigma})} \cdot \prod_{i=1}^{n} \Phi(\frac{y+\mu_i}{\sigma}) dy. \tag{3.4}$$

The details of obtaining Equation 3.4 are in Appendix A.1.

### 3.3.2 Discarding Mechanism

The next step is to decide, at the end of each round, the appropriate value of $k$, which determines how many $(n - k)$ lowest-ranked candidates will be discarded in this round. Our scheme decides $k$ based on the confidence curve: choosing the smallest $k$ that satisfies $P_k \geq \tau$, where $\tau$ is a parameter determined by our scheme adaptively as follows.

**Choosing $\tau$.** At the end of round $i$, we have the *confidence curve* $\mathcal{C}_i(P_1^i, P_2^i, \cdots, P_n^i)$ that is the trajectory of a series of probabilities. We quantify how $\tau$ influences the probability for the HPO to select the best candidate.

Let $\tau_i$ be the value of $\tau$ for round $i$, $k_i$ be $\min\{k : \mathcal{C}_i(P_k) \geq \tau_i\}$. As the scheme discards the worst $n - k_i$ candidates and further trains the best $k_i$ candidates in round $i + 1$, we can derive the *confidence curve* of round $i + 1$ as $\mathcal{C}_{i+1}(P_1^{i+1}, P_2^{i+1}, \cdots, P_{k_i}^{i+1})$ based on those selected $k_i$ candidates. Since we want to quantify the effect of $\tau$ on the probability that the scheme returns the best candidate (that is, to suppose round $i + 1$ is our final round), $P_1^{i+1}$ is the target we desire to maximize. Define $\boldsymbol{\xi}_i$ to be the current condition $(\mu_i, \sigma_i)_{i=1}^n$. Let $f(\cdot, \cdot)$ be a mapping such that $f(\boldsymbol{\xi}_i, \tau_i) = P_1^{i+1}$. We want to use a selector function $\Psi : D \to [0, 1]$ where $D = ([0, \infty) \times [0, \infty))^n \times [0, 1]$. $\Psi$ takes $\boldsymbol{\xi}_i$ as input and returns an optimal $\tau_i$:

$$\Psi(\boldsymbol{\xi}_i) = \arg \max_{\tau_i \in [0,1]} \{f(\boldsymbol{\xi}_i, \tau_i)\}. \tag{3.5}$$

The effect of $\tau_i$ on $f$ manifests through its influence on the number of candidates $k_i$ retained in the subsequent round, and can be ultimately broken down into the influence of (1) *exploration*, meaning keeping more candidates in the next round can reduce this round's discarding error, and (2) *exploitation*, meaning keeping fewer candidates in the next round can allow each candidate to receive more training time (recall that the training time budget is fixed for each round) and hence will increase the reliability of the validation at the end of the next round.

**Exploration.** If $k_i$ drops by 1 to $k_i'$, according to the definition of the *confidence curve*, the probability that the final optimal configuration is among the remaining candidates we keep drops by $\Delta c_{\downarrow} = P_{k_i} - P_{k_i'}$:

$$\Delta c_{\downarrow} = \int_{-\infty}^{\infty} \frac{1}{\sigma} \cdot \frac{\phi(\frac{y+\mu_{k_i}}{\sigma})}{\Phi(\frac{y+\mu_{k_i}}{\sigma})} \cdot \prod_{i=1}^{n} \Phi(\frac{y+\mu_i}{\sigma}) dy. \tag{3.6}$$

**Exploitation.** At the same time, a drop in $k_i$ leads to an increase in the individual training budget $b$. Let $\zeta$ be the coefficient that relates the increase in the number of training epochs to its corresponding effect on confidence. Using an approach similar to that employed in formulating the *confidence curve*,

$$\zeta = \int_{-\infty}^{\infty} \frac{1}{\sigma - \Delta_t \sigma} \phi(\frac{y+\mu_1}{\sigma - \Delta_t \sigma}) \cdot \prod_{i=2}^{k_i} \Phi(\frac{y+\mu_i}{\sigma - \Delta_t \sigma}) dy - \int_{-\infty}^{\infty} \frac{1}{\sigma} \phi(\frac{y+\mu_1}{\sigma}) \cdot \prod_{i=2}^{k_i} \Phi(\frac{y+\mu_i}{\sigma}) dy \tag{3.7}$$

where $t$ represents the current total number of epochs and $\Delta_t \sigma$ represents the reduction in the uncertainty $\sigma$ that would result from training each candidate for one additional epoch. The specifics for Equations 3.6 and 3.7 can be found in Appendix A.2. Given that $b$ increases by $\frac{R}{k_i'} - \frac{R}{k_i}$, the overall influence of *exploitation* on the probability of selecting an optimal candidate is $\Delta c_{\uparrow} = \frac{R}{k_i(k_i-1)} \zeta$.

Let $\zeta(k_i, \boldsymbol{\xi}_i)$ be the confidence increase, given condition $\boldsymbol{\xi}_i$, when each of the $k_i$ candidates gets a unit extra training budget. $\mathcal{C}_i(P_k), k \in [n]$ are the *confidence curves*. Balancing *exploration* and *exploitation* leads to a sweet point where $\Delta c_\downarrow = \Delta c_\uparrow$. That gives the way to derive the appropriate value for $\tau$, which just needs to make the following hold:

$$P_{k_i} - P_{k_i - 1} = \frac{R}{k_i(k_i - 1)} \zeta(k_i, \boldsymbol{\xi}_i). \tag{3.8}$$

## 3.4 Theoretical Analysis

We consider how the method performs in terms of identifying the best candidate. For convenience, we let $\ell_{i,t}$ be the approximation of the converged loss for the model with hyperparameter $\gamma_i$ at time $t$. For each $i$, assume $\nu_i = \lim_{\tau \to \infty} \ell_{i,\tau}$ exists. The goal is to identify $\arg\min_i \nu_i$. Without loss of generality, assume that $\nu_1 < \nu_2 \leq \cdots \leq \nu_n$. The assumption that $\lim_{\tau \to \infty} \ell_{i,\tau}$ exists implies that as $\tau$ grows, the overall gap between $\ell_{i,\tau}$ and $\nu_i$ decreases. Let $\sigma_t = f(t)$ be the model uncertainty at epoch $t$. We then introduce a random variable that characterizes the approximation error of $\ell_{i,t}$ relative to $\nu_i$, modeling it as a distribution that incorporates $t$ as a parameter:

$$X_t = \ell_{i,t} - \nu_i, X_t \sim \mathcal{N}(0, \sigma_t^2) \quad \forall t.$$

By applying Chebyshev inequality, we have

$$\Pr\left\{ |\ell_{i,t} - \nu_i| > \frac{\nu_i - \nu_1}{2} \right\} \leq \frac{4\sigma_t^2}{(\nu_i - \nu_1)^2} = \frac{4f(t)^2}{(\nu_i - \nu_1)^2} \quad i = 2, \cdots, n. \tag{3.9}$$

Let $\mathcal{A}$ denote the event $\ell_{i,t} > \ell_{1,t}$, then by Equation 3.9

$$\Pr(\mathcal{A}) = \Pr\left\{ (\ell_{i,t} - \nu_i) + (\nu_1 - \ell_{1,t}) + 2 \cdot (\frac{\nu_i - \nu_1}{2}) > 0 \right\} \geq 1 - \left( \frac{4f(t)^2}{(\nu_i - \nu_1)^2} \right)^2. \tag{3.10}$$

Equation 3.10 tells us that $\ell_{i,t} > \ell_{1,t}$ has a high probability with respect to $t$ if $f(t) \in O(t^{-1/4})$ (see Lemma in Appendix C). That is, comparing the intermediate values at a certain time $t$ is likely to establish an order similar to the order of the final values of $\nu_i$ and $\nu_1$.

The following theorem is stated using the abovementioned quantities with proofs in Appendix C.1.

**Theorem 1.** *Let $n$ be the number of total candidates, and $\nu_i = \lim_{\tau \to \infty} \ell_{i,\tau}$. For a given $c > 0$, there exists a $T > 0$ s.t. $\prod_{i=2}^{n} (1 - (\frac{4f(T)^2}{(\nu_i - \nu_1)^2})^2) > 1 - c$. If the round budget $R > T \cdot n$, then the best candidate is returned with probability $P > (1 - \lfloor \frac{B}{R} \rfloor c)(1 - c)$, where $B$ is the total budget.*

In comparison, the bound in the UQ-oblivious approach is as follows:

**Theorem 2.** *Let $\delta > 0, \nu_i = \lim_{\tau \to \infty} \ell_{i,\tau}$ and assume $\nu_1 \leq \nu_2 \leq \cdots \leq \nu_n$. Let $\gamma^{-1}(\epsilon, \delta) = \min\{t \in \mathbb{N} : \frac{f(t)}{\epsilon} \leq \delta^{\frac{1}{4}}\}$, and*

$$z_{ob} = 2\lceil \log_2(n) \rceil \max_{i=2,\dots,n} i \left( 1 + \gamma^{-1}(\frac{\nu_i - \nu_1}{2}, \delta) \right)$$

$$\leq 2\lceil \log_2(n) \rceil (n + \sum_{i=2,\dots,n} \gamma^{-1}(\frac{\nu_i - \nu_1}{2}, \delta)).$$

*If the UQ-oblivious early stopping method is run with any budget $B_{ob} > z_{ob}$ then the best candidate is returned with probability $P_{ob} > 1 - n\delta$.*

**Example 3.** *Consider $f(t) = \frac{1}{t}$. According to Theorem 2, if $B_{ob} > 2\lceil \log_2(n) \rceil (n + \sum_{i=1,\dots,n} \gamma^{-1}(\frac{\nu_i - \nu_1}{2}, \delta))$, the UQ-oblivious method can return the best candidate with probability over $1 - n\delta$. But if $B_{UQ} \simeq \gamma^{-1}(\frac{\nu_2 - \nu_1}{2}, \delta) \cdot n$ [1], the UQ method can return the best candidate with probability over $1 - n\delta$. As shown in Appendix C.2, Theorems 1 and 2 together show that the UQ approach guarantees the same probability of identifying the optimal candidate as the UQ-oblivious counterpart with a smaller budget lowerbound $B$ (see Corollary 6).*

---

[1] $f \simeq g$ if there are constants $c, c'$ s.t. $cg(x) \leq f(x) \leq c'g(x)$.

### 3.5 UQ-Guided HPO Family

The *UQ-guided scheme* is a general approach to enhancing HPO with uncertainty awareness. We next explain how it is integrated into several existing HPO methods to transform them into UQ-guided ones, yielding a UQ-guided HPO family. In the following, we use the suffix "plus (+)" to indicate the UQ-guided HPO methods.

**Successive Halving plus (SH+)** is derived from the early stop-based HPO design Successive Halving (SH) [19]. Algorithms 1 and 2 in Appendix A.3 show the pseudo-code. Given total budget $B$ and round budget $R$ and an initial $K$, SH+ first trains $K$ candidates each with the initial $b = \lfloor \frac{R}{K} \rfloor$ units of budget, and ranks them by the evaluation performance. Then SH+ updates $K$ based on Section 3.3.2 and keeps the top $K$ configurations according to the UQ-guided scheme ($OracleModel$ in Algorithms 1 and 2), and continues the process until the budget runs out.

**Hyperband plus (HB+)** originates from the popular HPO design Hyperband (HB). HB is an HPO method trying to better balance exploration and exploitation than SH does [25] by adding an outer loop for grid search of the value of $K$. HB+ simply extends HB by using SH+ rather than SH as its inner loop, changing the target of the grid search to the initial value of $K$.

**Bayesian Optimization and Hyperband plus (BOHB+)** is developed from BOHB [11]. BOHB is similar to HB except that it replaces the random sampling from the uniform distribution with BO-based sampling. BOHB+ makes the corresponding changes from HB+ by adopting BO-based sampling for its outer loop.

**Sub-sampling plus (SS+)** is derived from the Sub-sampling (SS) algorithm [15]. It showcases the applicability of the UQ-guided scheme to non-early stop–based methods. Similar to other methods, in each round, SS also chooses candidates for further training based on its assessment of the potential of those candidates. But unlike the other methods, SS does not discard any candidates, but keeps all in play throughout the entire HPO process. In each round, the candidates it chooses are those that show smaller validation loss than the most trained candidate shows. If there is none, it trains only the most trained candidate in that round. SS+ integrates the UQ-guided scheme into the candidate selection process of SS. When SS+ compares a candidate ($c_i$) against the most trained candidate ($c_m$), rather than checking their validation losses, it uses the UQ-guided scheme to compute the probability for the convergence loss of $c_i$ to be smaller than that of $c_m$ and checks whether the probability is over a threshold $\tau$ (0.9 in our experiments), that is, $\Pr(\ell(y, M^*_{\gamma_{c_m}}(\mathbf{x})) \geq \ell(y, M^*_{\gamma_{c_i}}(\mathbf{x}))) \geq \tau$.

## 4 Experiments

We conduct a series of experiments on the four UQ-guided HPO methods to validate the efficacy of the UQ-guided scheme for HPO.

### 4.1 Experimental Setup

**Methodology.** To check the benefits of the UQ-guided scheme for HPO, we apply the proposed UQ-guided HPO family to different HPO benchmarks, including NAS-BENCH-201 and LCBench, each for 30 repetitions, to measure the performance for different hyperparameter optimization tasks, and compared those with their original UQ-oblivious versions.

**Platform.** Our experiments are conducted on a platform equipped with an Intel i9-9900k CPU and an NVIDIA GEFORCE RTX 2080 TI GPU. The CPU has 8 cores, each of which can support 2 threads. The GPU has 4,352 cores of Turing architecture with a computing capability of 7.5. The GPU can achieve a maximum memory bandwidth of 616 GB/s, 0.4 tera floating-point operations per second (TFLOPS) on double-precision, and 13 TFLOPS on single-precision.

**Workloads.** We evaluate the UQ-guided methods on two real-world benchmarks. Nas-Bench-201 [9] (CC-BY 4.0) encompasses three heavyweight neural architecture search tasks (NAS) on CIFAR-10, CIFAR-100, and ImageNet-16-12 (CC-BY 4.0) datasets. In addition, we investigate the performance of optimizing traditional ML pipelines, hyperparameters, and neural architecture in LCBench [42]. For example, we optimized 7 parameters for the Fashion-MNIST dataset [7], where the resource type is determined by the number of iterations. Additional information regarding these benchmarks can be found in Appendix F. In this context, one unit of budget equates to a single training epoch, and by default, the total HPO budget ($B$) allocated for each method is 4 hours.

### 4.2 Experimental Results

Figure 3 illustrates the results of NAS-BENCH-201 trained on ImageNet-16-120. It shows the results of four UQ-guided methods compared to their original ones. For each comparison, we show three

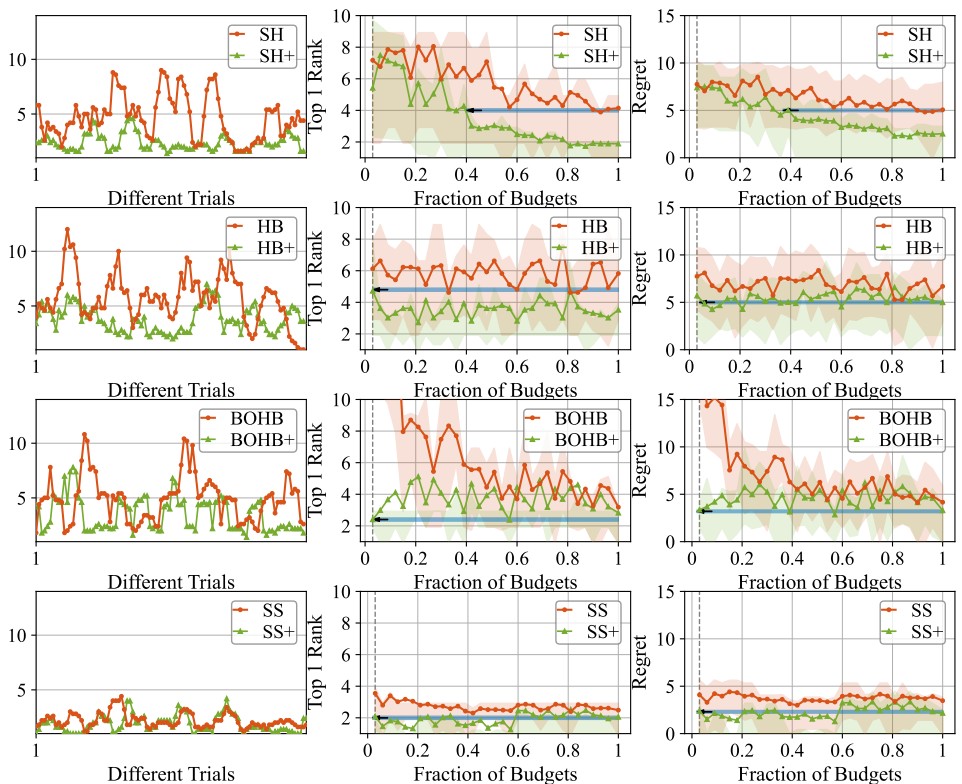

Figure 3: Experimental results of UQ-oblivious HPO methods and their UQ-guided enhancements on NAS-BENCH-2.0.

metrics, namely top-1 rank on different trials, top-1 rank on different fractions of budgets, and regret on different fractions of budgets. The fraction of budgets denotes the portion of the budget that we allocate for that particular experiment compared to the standard full budget. Top-1 rank refers to the real ranking of the candidate ultimately chosen by the method. Regret (%) refers to the accuracy difference between the returned candidate and the real best candidate. In Figure 3, the average results of 30 repetitions are reported. For the right two columns, we also report the uncertainty bands, defined as the interval between the 30th and 70th percentiles. The benefits of the UQ-guided scheme are obvious, both for individual trials and across different fractions of budgets. It brings a 21-55% regret reduction. Similar results are observed on other benchmarks (LCBench results shown in Appendix G).

Table 1 provides the fraction of the total exploration time needed for the UQ-guided methods to achieve comparable model accuracy as the original methods do. The UQ-guided methods need much less time than their counterparts to obtain a similar performance. For instance, SH+ achieves the same average regret of 5% on NAS with only half of the budgets required by SH. These results indicate that the UQ technique can conduct HPO efficiently and effectively.

Table 1: Fraction of time (%) required for the UQ-guided methods to achieve comparable model performance as the original HPO methods do.

| Methods | NAS-BENCH-201 | LCBench |
|---------|---------------|---------|
| SH+     | 50.78         | 43      |
| HB+     | 75            | 60      |
| BOHB+   | 68.4          | 53.34   |
| SS+     | 47.84         | 30.93   |

Our paper's experiments concentrate on DNN because efficient HPO is crucial for the time-consuming nature of DNN training. We also use the ridge regression in Section H as a demonstration to show the

potential of the methodology for other iterative learners, and leave a systematic study of which to the future.

## 5 Conclusion

This paper points out the importance of systematic treatment to the uncertainty in model trainings for HPO. It introduces a novel scheme named *UQ-guided scheme*, which offers a general way to enhance HPO methods for DNNs with uncertainty awareness. Experiments demonstrate that the UQ-guided scheme can be easily integrated into various HPO methods. The enhanced methods achieve 21–55% reduction of regret over their original versions, and require only 30–75% time to identify a candidate with a matching performance as the original methods do. The paper in addition provides a theoretical analysis of the effects of the UQ-guided scheme for HPO.

The key characteristic of the UQ method is the necessity to rank multiple learners during the HPO process. Gradient-based HPO methods [30], for instance, may not benefit from our UQ-guided scheme because of their sequential properties. One limitation of this paper is that it is mostly suitable for iterative learners, and needs adaptations for other learners: To go beyond, it could be, for instance, applied to the model selection work in previous studies [31] that use training dataset size as the budget dimension. In this case, the learner does not need to be iterative; the selection is based on the validation loss history trained with incremental dataset sizes. The UQ component can still guide the configuration selection and budget allocation in the HPO process.

Overall, this study concludes that UQ is important for HPO to consider, simple on-the-fly UQ goes a long way for HPO, and the *UQ-guided scheme* can serve as a general effective scheme for enhancing HPO designs.

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

# A  Method Details

## A.1  Formulation of *Confidence Curve*

The following contents detail how to compute the *confidence curve* $\mathcal{C}(P_1, P_2, \cdots, P_n)$ based on current validation loss and quantified uncertainty of the $n$ candidates.

Let $Y$ be the random variable denoting the negation of the lowest converged validation loss:

$$Y = \max(-\ell(\mathbf{y}, M^*_{\gamma_1}(\mathbf{X})), -\ell(\mathbf{y}, M^*_{\gamma_2}(\mathbf{X})), \cdots, -\ell(\mathbf{y}, M^*_{\gamma_n}(\mathbf{X}))).$$

Since $\ell(\mathbf{y}, M^*_{\gamma_i}(\mathbf{X})) \sim \mathcal{N}(\mu_i, \sigma_i^2)$, the cumulative distribution function (CDF) of $Y$, $F_Y(y)$, is

$$F_Y(y) = \Pr(Y \le y) = \Pr(-\ell(\mathbf{y}, M^*_{\gamma_1}(\mathbf{X})) \le y, -\ell(\mathbf{y}, M^*_{\gamma_2}(\mathbf{X})) \le y, \cdots, -\ell(\mathbf{y}, M^*_{\gamma_n}(\mathbf{X})) \le y)$$

$$= \prod_{i=1}^{n} \Phi(\frac{y + \mu_i}{\sigma}) = \exp(\sum_{i=1}^{n} \ln \Phi(\frac{y + \mu_i}{\sigma})).$$

Accordingly, the probability density function (PDF) of $Y$, $f_Y(y)$, is

$$f_Y(y) = \frac{dF_Y(y)}{dy} = \frac{1}{\sigma} \sum_{i=1}^{n} \frac{\phi(\frac{y+\mu_i}{\sigma})}{\Phi(\frac{y+\mu_i}{\sigma})} \cdot \prod_{i=1}^{n} \Phi(\frac{y + \mu_i}{\sigma}).$$

Now we can construct the *confidence curve* by calculating each $P_k$ ($k \in [n]$). For $k = m$, $P_k$ can be expressed as

$$P_m = \Pr(\min(\ell(\mathbf{y}, M^*_{\gamma_1}(\mathbf{X})), \cdots, \ell(\mathbf{y}, M^*_{\gamma_m}(\mathbf{X}))) \le \min(\ell(\mathbf{y}, M^*_{\gamma_{m+1}}(\mathbf{X})), \cdots, \ell(\mathbf{y}, M^*_{\gamma_n}(\mathbf{X})))).$$

Let $\ell(\mathbf{y}, M^*_{\gamma_1}(\mathbf{X})), \cdots, \ell(\mathbf{y}, M^*_{\gamma_n}(\mathbf{X}))$ be mutually independent, thus

$$
\begin{aligned}
P_m &= \int_{-\infty}^{\infty} f_Y(y) \Pr(-\mathcal{N}(\mu_{m+1}, \sigma_{m+1}) \le y, \cdots, -\mathcal{N}(\mu_n, \sigma_n) \le y) dy \\
&= \int_{-\infty}^{\infty} f_Y(y) \Phi(\frac{y + \mu_{m+1}}{\sigma}) \times \cdots \times \Phi(\frac{y + \mu_n}{\sigma}) dy \\
&= \int_{-\infty}^{\infty} \frac{1}{\sigma} \sum_{i=1}^{m} \frac{\phi(\frac{y+\mu_i}{\sigma})}{\Phi(\frac{y+\mu_i}{\sigma})} \cdot \prod_{i=1}^{n} \Phi(\frac{y + \mu_i}{\sigma}) dy.
\end{aligned}
\tag{A.1}
$$

## A.2  Computing $\zeta$

Equation 3.6 can be calculated by directly subtracting $P_{k_i}$ by $P_{k_i - 1}$:

$$
\begin{aligned}
\Delta c_{\downarrow} &= \int_{-\infty}^{\infty} \frac{1}{\sigma} \sum_{i=1}^{k_i} \frac{\phi(\frac{y+\mu_i}{\sigma})}{\Phi(\frac{y+\mu_i}{\sigma})} \cdot \prod_{i=1}^{n} \Phi(\frac{y + \mu_i}{\sigma}) dy - \int_{-\infty}^{\infty} \frac{1}{\sigma} \sum_{i=1}^{k_i-1} \frac{\phi(\frac{y+\mu_i}{\sigma})}{\Phi(\frac{y+\mu_i}{\sigma})} \cdot \prod_{i=1}^{n} \Phi(\frac{y + \mu_i}{\sigma}) dy \\
&= \int_{-\infty}^{\infty} \frac{1}{\sigma} \cdot \frac{\phi(\frac{y+\mu_{k_i}}{\sigma})}{\Phi(\frac{y+\mu_{k_i}}{\sigma})} \cdot \prod_{i=1}^{n} \Phi(\frac{y + \mu_i}{\sigma}) dy.
\end{aligned}
$$

$\zeta$ is the coefficient that relates the increase in the number of training epochs to its corresponding effect on confidence. Consider a working round that starts with $k$ candidates. We have approximations at the end of the round for the converged loss $\ell(y, M^*_{\gamma_i}(\mathbf{x})) \sim \mathcal{N}(\mu_i, \sigma^2)$ for $i \in [k]$. Here, $t$ denotes the epochs. Letting each candidate train one extra unit of resource results in lower uncertainty, thus increasing the $f$ score. First compute $f(\boldsymbol{\xi}, \tau)$ at epoch $t$:

$$f(\boldsymbol{\xi}, \tau)_t = \int_{-\infty}^{\infty} \frac{1}{\sigma} \phi(\frac{y + \mu_1}{\sigma}) \cdot \prod_{i=2}^{k} \Phi(\frac{y + \mu_i}{\sigma}) dy. \tag{A.2}$$

If each configuration is trained with one extra unit of resource (a total of $t + 1$ epochs), model uncertainty would be reduced. We use the same $\mu_i$ to approximate the converged validation loss

for $t + 1$ epochs, and use $\sigma - \Delta_t\sigma$ as an approximation for model uncertainty. Here $\Delta_t\sigma$ is the decrease in model uncertainty that is determined through offline profiling with details in Section D. This approximation leads us to $f(\boldsymbol{\xi}, \tau)$ at epoch $t + 1$ as

$$f(\boldsymbol{\xi}, \tau)_{t+1} = \int_{-\infty}^{\infty} \frac{1}{\sigma - \Delta_t\sigma}\phi(\frac{y + \mu_1}{\sigma - \Delta_t\sigma}) \cdot \prod_{i=2}^{k} \Phi(\frac{y + \mu_i}{\sigma - \Delta_t\sigma})dy. \tag{A.3}$$

Subtracting Equation A.3 by Equation A.2 gives us the result in Equation 3.7:

$$\zeta = f(\boldsymbol{\xi}, \tau)_{t+1} - f(\boldsymbol{\xi}, \tau)_t = \int_{-\infty}^{\infty} \frac{1}{\sigma - \Delta_t\sigma}\phi(\frac{y + \mu_1}{\sigma - \Delta_t\sigma}) \cdot \prod_{i=2}^{k} \Phi(\frac{y + \mu_i}{\sigma - \Delta_t\sigma})dy - \int_{-\infty}^{\infty} \frac{1}{\sigma}\phi(\frac{y + \mu_1}{\sigma}) \cdot \prod_{i=2}^{k} \Phi(\frac{y + \mu_i}{\sigma})dy.$$

### A.3 Pseudo-code for UQ-Guided Hyperparameter Optimization (SH+)

---

**Algorithm 1** UQ-Guided Hyperparameter Optimization (SH+)

---

**Input:** Total budget $B$, the set of $K$ configurations $\Gamma = \{\gamma_1, \gamma_2, \cdots, \gamma_K\}$, minimum round budget $R$

**Output:** The configuration with the best performance

$b = \lfloor \frac{R}{K} \rceil$

**repeat**
  **for** $i = 1$ **to** $K$ **do**
    Evaluate $M_{\gamma_i}^{t_i}$ with budget $b$ and get $M_{\gamma_i}^{t_i + b}$
    $t_i += b$
  **end for**
  Rank according to performance and obtain new $M_{\gamma_1}^{t_1}, M_{\gamma_2}^{t_2}, \cdots, M_{\gamma_K}^{t_K}$
  $K = OracleModel(M_{\gamma_1}^{t_1}, M_{\gamma_2}^{t_2}, \cdots, M_{\gamma_K}^{t_K})$
  Keep top $K$ candidates
**until** total budget $B$ runs out

---

---

**Algorithm 2** $OracleModel$ for determining $K$ candidates into the next round.

---

**Input:** $K$ instances of Machine Learning models $M_{\gamma_1}^{t_1}, M_{\gamma_2}^{t_2}, \cdots, M_{\gamma_K}^{t_K}$
**Output:** A new $K$ that tells the model how many candidates to keep
Get a $\tau$ from the probabilistic model
Construct the *confidence curve* $\mathcal{C}(P_1, P_2, \cdots, P_K)$ based on $\ell(\mathbf{y}, M_{\gamma_1}^{t_1}(\mathbf{X})), \ell(\mathbf{y}, M_{\gamma_2}^{t_2}(\mathbf{X})), \cdots, \ell(\mathbf{y}, M_{\gamma_K}^{t_K}(\mathbf{X}))$
**return** $\min\{k : P_k > \tau\}$

---

### A.4 Pseudo-code for UQ-Guided Hyperparameter Optimization (HB+)

---

**Algorithm 3** Hyperband plus (HB+)

---

**Input:** Total budget $B$, the set of $K$ configurations $\Gamma = \{\gamma_1, \gamma_2, \cdots, \gamma_K\}$, maximum budget $R$, ratio $\eta$
**Output:** The configuration with the best performance
$s_{\max} = \lfloor \log_\eta R \rfloor, b = \frac{B}{s_{\max}}$
**for** $s = 1$ **to** $s_{\max}$ **do**
  $k = \lceil \frac{b\eta^s}{s+1} \rceil$
  Sample $k$ configurations randomly
  Call SH+ with $(k, b, \frac{b}{s})$
**end for**

---

### A.5 Pseudo-code for UQ-Guided Hyperparameter Optimization (BOHB+)

---

**Algorithm 4** Bayesian Optimization Hyperband plus (BOHB+)

---

**Input:** Total budget $B$, the set of $K$ configurations $\Gamma = \{\gamma_1, \gamma_2, \cdots, \gamma_K\}$, maximum budget $R$, ratio $\eta$ (default $\eta = 3$)
**Output:** The configuration with the best performance
$s_{\max} = \lfloor \log_\eta R \rfloor, b = \frac{B}{s_{\max}}$
**for** $s = 1$ **to** $s_{\max}$ **do**
   $k = \lceil \frac{b\eta^s}{s+1} \rceil$
   Sample $k$ configurations using Bayesian optimizer
   Call SH+ with $(k, b, \frac{b}{s})$
**end for**

---

## A.6    Pseudo-code for UQ-Guided Hyperparameter Optimization (SS+)

---

**Algorithm 5** Sub-Sampling plus (SS+)

---

**Input:** Total budget $B$, the set of $K$ configurations $\Gamma = \{\gamma_1, \gamma_2, \cdots, \gamma_K\}$, maximum budget $R$, minimum budget $b$, ratio $\eta$ (default $\eta = 3$)
**Output:** The configuration with the best performance
$r = 1$, evaluate all configurations with budget $b$.
**for** $r = 2$ **to** $\lfloor \log_\eta (R/b) \rfloor$ **do**
   Select $\gamma_\zeta$ with the most observations.
   $\mathcal{I}' = \{k : c_k \in \Gamma \backslash \gamma_\zeta, \gamma_k \preceq \gamma_\zeta \wedge \text{pass UQ check}\}$
   **if** $\mathcal{I}' == \emptyset$ **then**
      Evaluate $\gamma_\zeta$ with budget $\eta^r b$.
   **else**
      Evaluate $\gamma_k$ with budget $\eta^r b$ for each $k \in \mathcal{I}'$.
   **end if**
**end for**

---

# B    More Theoretical Analysis

We have been concerned with identifying the best candidate, while in practice, it is often sufficient to consider a situation where the difference between the result of candidate $i_\epsilon$ ($\nu_{i_\epsilon}$) and the result of the best candidate ($\nu_1$) is less than or equal to a small value $\epsilon$. We obtain the following theorem with proofs in Appendix C.3.

**Theorem 4.** *For a budget $B > R$ and a set of $n$ candidates, let $\hat{i}$ be the output of the UQ-guided approach. Then*

$$\mathbb{E}(\nu_{\hat{i}} - \nu_1) \leq \frac{2\lfloor \frac{B}{R} \rfloor \sqrt{2} f(R)}{\sqrt{\pi}}.$$

*In comparison, $\hat{i}_D$, the output of the UQ-oblivious counterpart, satisfies*

$$\mathbb{E}(\nu_{\hat{i}_D} - \nu_1) \leq \frac{2\lceil \log_2(n) \rceil \sqrt{2} f(\lfloor \frac{B}{n\lceil \log_2(n) \rceil} \rfloor)}{\sqrt{\pi}}.$$

**Example 5.** *Consider $f(t) = \frac{1}{t}$. Substitution of $f(t)$ in Theorem 4 can clearly show a smaller upperbound of the UQ-guided approach than that of the UQ-oblivious counterpart (see Appendix C.3 for details).*

The theorems provide some insights into the theoretical benefits of the UQ-guided scheme. But it is worth noting that neither this bound comparison nor the budget bound comparison in Example 3 is sufficient to prove that the UQ-guided approach definitely would outperform the UQ-oblivious approach, a reason for the empirical comparisons in Section 4.

## C Proofs

In this section, we provide proofs for the theorems presented in Section 3.4. At the end of the proof, we let $f(t) = \frac{1}{t}$ and obtained the results in Example 3 and Example 5.

The Lemma stated next will prove to be useful.

**Lemma 1.** *For $i > 1$, if $\min\{t_1, t_i\} > t$, then we have a high probability that $\ell_{i,t_i} > \ell_{1,t_1}$ with respect to $t$ if $f(t) \in O(t^{-1/4})$.*

*Proof.* In Section 3.4, we come to the conclusion that

$$\Pr(\ell_{i,t} > \ell_{1,t}) \geq 1 - (\frac{4f(t)^2}{(\nu_i - \nu_1)^2})^2 = 1 - (\frac{2}{\nu_i - \nu_1})^4 \cdot f(t)^4 > 1 - O\left(\frac{1}{t}\right).$$

This shows that the event $\ell_{i,t} > \ell_{1,t}$ happens with high probability with respect to $t$.

Now consider a more general setting, where each $\ell_i$ has its own $t_i$:

$$\Pr(|\ell_{i,t_i} - \nu_i| > \frac{\nu_i - \nu_1}{2}) \leq \frac{4f(t_i)^2}{(\nu_i - \nu_1)^2} \quad i = 1, \cdots, n.$$

Comparing $\ell_{i,t_i}$ and $\ell_{1,t_1}$ for a particular $i \in [n]$ gives us the following:

$$\Pr(\ell_{i,t_i} > \ell_{1,t_1}) = \Pr((\ell_{i,t_i} - \nu_i) + (\nu_1 - \ell_{1,t_1}) + 2 \cdot \frac{\nu_i - \nu_1}{2} > 0) \geq 1 - \frac{4f(t_1)^2}{(\nu_i - \nu_1)^2} \cdot \frac{4f(t_i)^2}{(\nu_i - \nu_1)^2}. \tag{C.1}$$

Since $t_1 > t$ and $t_i > t$,

$$(C.1) > 1 - O\left(\frac{1}{t}\right).$$

$\square$

### C.1 Proof of Theorem 1

*Proof.* Let $S_i$ be the set of candidates the UQ scheme evaluates at the beginning of the $i$-th round. We assume that the $n$ infinitely long loss sequences $[\ell_{i,t}]$ with limits $\{\nu_i\}_{i=1}^n$.

We compute the probability that the algorithm includes the best candidate in the last round, namely, $1 \in S_{\lfloor \frac{B}{R} \rfloor}$, and the probability that the UQ scheme returns the best candidate in $S_{\lfloor \frac{B}{R} \rfloor}$.

Let $r_k$ be the round budget for each candidate in $S_k$. $R_k = \sum_{j=0}^k r_k$. The probability that the best candidate is among the final kept candidate set is

$$\begin{aligned}
\Pr(1 \in S_{\lfloor \frac{B}{R} \rfloor}) &= 1 - \sum_{k=1}^{r=\lfloor \frac{B}{R} \rfloor} \Pr(1 \notin S_k, 1 \in S_{k-1}) \\
&= 1 - \sum_{k=0}^{r=\lfloor \frac{B}{R} \rfloor - 1} (\Pr(1 \in S_{k-1}) - \Pr(1 \in S_k, 1 \in S_{k-1})) \qquad \text{(C.2)} \\
&\geq 1 - \sum_{k=0}^{r=\lfloor \frac{B}{R} \rfloor - 1} \left( 1 - \Pr\left( \bigwedge_{i \in S_k \setminus \{1\}} \ell_{i,R_k} > \ell_{1,R_k} \right) \right).
\end{aligned}$$

Since the probability that $\ell_{1,t}$ is the smallest among all $\ell_{k,t}$ ($k \in [n]$) is greater than

$$\prod_{i=2}^n \Pr(\ell_{i,t} > \ell_{1,t}) \geq \prod_{i=2}^n (1 - \left( \frac{4f(t)^2}{(\nu_i - \nu_1)^2} \right)^2),$$

we have

$$(C.2) \geq 1 - \sum_{k=0}^{\lfloor \frac{B}{R} \rfloor - 1} (1 - \prod_{i=2}^{n} (1 - (\frac{4f(R_k)^2}{(\nu_i - \nu_1)^2})^2))$$

$$= 1 - \sum_{k=0}^{\lfloor \frac{B}{R} \rfloor - 1} (1 - \prod_{i=2}^{n} (1 - (\frac{4f(\sum_{j=0}^{k} \frac{R}{|S_j|})^2}{(\nu_i - \nu_1)^2})^2))$$

$$\geq 1 - \lfloor \frac{B}{R} \rfloor c.$$

Therefore, the probability that the scheme returns the best candidate is no less than

$$\Pr(1 \in S_{\lfloor \frac{B}{R} \rfloor}) \cdot \Pr\left( \bigwedge_{i \in S_{\lfloor \frac{B}{R} \rfloor}} \ell_{i, R_{\lfloor \frac{B}{R} \rfloor}} > \ell_{1, R_{\lfloor \frac{B}{R} \rfloor}} \right) \geq (1 - \lfloor \frac{B}{R} \rfloor c)(1 - c). \tag{C.3}$$

$\square$

That is to say, for a given confidence threshold $c$, there exists a $T$ such that as long as $\min\{t_1, t_2, ..., t_n\} > T$, then $\Pr(\wedge_{i=2,...,n}(\ell_{i,t_i} > \ell_{1,t_1})) > 1 - c$. Recall that in the UQ scheme design, the goal is to select an optimal hyperparameter configuration from $n$ candidates, and each round is allocated for $R$ resources. This means that if choose $R \geq T \cdot n$, then the best candidate is returned from the algorithm with probability $P > (1 - \lfloor \frac{B}{R} \rfloor c)(1 - c)$.

## C.2 Proof of Theorem 2

The representation of $z_{ob}$ on the right-hand-side of the inequality is very intuitive: For each $i$, to confirm that the final loss of the $i$-th candidate is greater than the best candidate's with a probability of at least $1 - \delta$, it is necessary to train both candidates for at least the number of steps indicated by the $i$-th term in the sum. Repeating this reasoning for all $i$ justifies the sum over all candidates.

*Proof.* First we show that, given budget $z = z_{ob}$, round budget for round $k$ satisfies

$$r_k \geq \frac{z}{|S_k| \lceil \log_2 n \rceil} - 1$$

$$= \frac{2}{|S_k|} \max_{i=2,...,n} i \left(1 + \gamma^{-1}(\frac{\nu_i - \nu_1}{2}, \delta)\right) - 1$$

$$\geq \frac{2}{|S_k|} (\lfloor |S_k|/2 \rfloor + 1) \left(1 + \gamma^{-1}(\frac{\nu_{\lfloor |S_k|/2 \rfloor + 1} - \nu_1}{2}, \delta)\right) - 1$$

$$\geq \gamma^{-1}(\frac{\nu_{\lfloor |S_k|/2 \rfloor + 1} - \nu_1}{2}, \delta).$$

The last inequality is derived because $\lfloor |S_k|/2 \rfloor \geq |S_k|/2 - 1$.

Let $\tau_i := \gamma^{-1}(\frac{\nu_i - \nu_1}{2}, \delta)$. We then show that, for a time $t$:

$$t \geq \tau_i \Rightarrow t \geq \gamma^{-1}(\frac{\nu_i - \nu_1}{2}, \delta)$$

$$\Leftrightarrow 1 - \left(\frac{4f(t)^2}{(\nu_i - \nu_1)^2}\right)^2 \geq 1 - \delta$$

$$\Rightarrow \Pr(\ell_{i,t} > \ell_{1,t}) \geq 1 - \delta.$$

The second line follows by the definition of $\gamma^{-1}(\epsilon, \delta)$. Since $r_k \geq \tau_{\lfloor |S_k|/2 \rfloor + 1}$, we can compute the following probability

$$\Pr(1 \in S_{k+1} | 1 \in S_k) = \Pr\left( \sum_{i \in S_k} \mathbf{1}\{\ell_{i,R_k} > \ell_{1,R_k}\} \geq \lfloor |S_k|/2 \rfloor \right)$$

$$\geq \Pr(\sum_{i=\lfloor |S_k|/2 \rfloor + 1}^{|S_k|} \mathbf{1}\{\ell_{i,R_k} > \ell_{1,R_k}\} \geq \lfloor |S_k|/2 \rfloor)$$

$$\geq (1 - \delta)^{\lfloor \frac{|S_k|}{2} \rfloor}$$

where the first line follows by the definition of the early stopping algorithm (Successive Halving), the second by $\tau_i$ being non-increasing. Namely, for all $i > \lfloor |S_k|/2 \rfloor + 1$, we have $\tau_i \leq \tau_{\lceil |S_k|/2 \rceil + 1}$ and consequently, $\Pr(\ell_{i,R_k} > \ell_{1,R_k}) \geq \Pr(\ell_{\lfloor |S_k|/2 \rfloor, R_k} > \ell_{1,R_k}) \geq 1 - \delta$.

Consequently, the probability that the UQ-oblivious approach returns the optimal candidate is

$$P_{ob} \geq \Pr \Big( \bigwedge_{i=0,\ldots,\lceil \log_2 n \rceil - 1} (1 \in S_{k+1} | 1 \in S_k) \Big)$$

$$\geq \prod_{k=0}^{\lceil \log_2 n \rceil - 1} (1 - \delta)^{\lfloor \frac{|S_k|}{2} \rfloor}$$

$$\geq 1 - n\delta.$$

We show in the next Corollary that, for $c = \frac{n \cdot \delta}{2}$, the probability $P$ obtained in Theorem 1 is no less than $1 - n\delta$. $\qquad \square$

**Corollary 6.** *For the threshold $c$ in Theorem 1 and $\delta$ in Theorem 2, let $c = \frac{n \cdot \delta}{2}$ and*
$$\beta^{-1}(\epsilon_2, \epsilon_3, \cdots, \epsilon_n, c) = \min\{T : \prod_{i=2}^{n}(1 - (\frac{f(T)}{\epsilon_i})^4) \geq 1 - c\} . \text{ Then by Theorem 1 the UQ}$$
*approach returns the best candidate with probability over $1 - n\delta$ if $B \simeq \gamma^{-1}(\frac{\nu_2 - \nu_1}{2}, \delta) \cdot n$.*

*Proof.* Let $T > \sqrt[4]{2} \cdot \gamma^{-1}(\frac{\nu_2 - \nu_1}{2}, \delta)$, we have

$$\prod_{i=2}^{n}(1 - (\frac{4f(T)^2}{(\nu_i - \nu_1)^2})^2) > (1 - \frac{\delta}{2})^n \geq 1 - \frac{n\delta}{2} = 1 - c.$$

This shows us that $T > \beta^{-1}(\frac{\nu_2 - \nu_1}{2}, ..., \frac{\nu_n - \nu_1}{2}, c)$. Consequently, according to Theorem 1, for $B = R > T \cdot n$, the UQ approach returns the best candidate with probability

$$(1 - c)^2 \geq 1 - 2c = 1 - n\delta.$$

The UQ-oblivious approach returns the optimal candidate with probability over $1 - n\delta$ if the budget $B_{ob} > z_{ob}$. But the UQ approach achieves the guarantee with budget $B > \sqrt[4]{2} \cdot \gamma^{-1}(\frac{\nu_2 - \nu_1}{2}, \delta) \cdot n$, which can be empirically substantially smaller than the budget required in Theorem 2.

$\qquad \square$

### C.3 Proof of Theorem 4

*Proof.* Let $R_k = \sum_{j=0}^{k} r_k$, namely, the total number of epochs allocated for each candidate in $S_k$. We can guarantee that, for the UQ-oblivious approach, the output candidate $\hat{i}_D$ satisfies

$$\mathbb{E}(\nu_{\hat{i}_D} - \nu_1) = \mathbb{E}\Big( \min_{i \in S_{\lceil \log_2(n) \rceil}} \nu_i - \nu_1 \Big)$$

$$= \mathbb{E}\Big( \sum_{k=0}^{\lceil \log_2(n) \rceil - 1} \min_{i \in S_{k+1}} \nu_i - \min_{i \in S_k} \nu_i \Big)$$

$$\leq \mathbb{E}\Big( \sum_{k=0}^{\lceil \log_2(n) \rceil - 1} 2|\nu_i - \ell_{i,R_k}| + \min_{i \in S_{k+1}} \ell_{i,R_k} - \min_{i \in S_k} \ell_{i,R_k} \Big)$$

$$= \mathbb{E}\Big( \sum_{k=0}^{\lceil \log_2(n) \rceil - 1} 2|\nu_i - \ell_{i,R_k}| \Big)$$

$$\leq \frac{2\lceil \log_2(n) \rceil \sqrt{2} f(\lfloor \frac{B}{n \lceil \log_2(n) \rceil} \rfloor)}{\sqrt{\pi}} = \frac{2\sqrt{2}n \lceil \log_2(n) \rceil^2}{\sqrt{\pi} B}.$$

by inspecting how the approach eliminates candidates and plugging in an upper bound for $\mathbb{E}(2|\nu_i - \ell_{i,R_k}|)$ for all $k$ in the last inequality. We can calculate the bound for the UQ-guided method in a

similar way:

$$
\begin{aligned}
\mathbb{E}(\nu_{\hat{i}} - \nu_1) &= \mathbb{E}\Big( \min_{i \in S_{\lfloor \frac{B}{R} \rfloor}} \nu_i - \nu_1 \Big) \\
&= \mathbb{E}\Big( \sum_{k=0}^{\lfloor \frac{B}{R} \rfloor - 1} \min_{i \in S_{k+1}} \nu_i - \min_{i \in S_k} \nu_i \Big) \\
&\leq \mathbb{E}\Big( \sum_{k=0}^{\lfloor \frac{B}{R} \rfloor - 1} \min_{i \in S_{k+1}} (\nu_i - \ell_{i,R_{k+1}} + \ell_{i,R_{k+1}}) - \min_{i \in S_k}(\nu_i - \ell_{i,R_k} + \ell_{i,R_k}) \Big) \\
&\leq \mathbb{E}\Big( \sum_{k=0}^{\lfloor \frac{B}{R} \rfloor - 1} 2|\nu_i - \ell_{i,R_k}| + \min_{i \in S_{k+1}} \ell_{i,R_{k+1}} - \min_{i \in S_k} \ell_{i,R_k} \Big) \\
&\leq \mathbb{E}\Big( \sum_{k=0}^{\lfloor \frac{B}{R} \rfloor - 1} 2|\nu_i - \ell_{i,R_k}| + (\min_{i \in S_{k+1}} \ell_{i,R_{k+1}} - \min_{i \in S_{k+1}} \ell_{i,R_k}) + (\min_{i \in S_{k+1}} \ell_{i,R_k} - \min_{i \in S_k} \ell_{i,R_k}) \Big) \\
&\leq \mathbb{E}\Big( \sum_{k=0}^{\lfloor \frac{B}{R} \rfloor - 1} 2|\nu_i - \ell_{i,R_k}| \Big) \\
&\leq \frac{2\lfloor \frac{B}{R} \rfloor \sqrt{2} f(R)}{\sqrt{\pi}} = \frac{2\sqrt{2}\lfloor \frac{B}{R} \rfloor}{\sqrt{\pi}R}.
\end{aligned}
$$

$\square$

**The smaller upperbound of the UQ-guided approach than that of the UQ-oblivious counterpart in Theorem 4.**

A simple calculation reveals that

$$
\frac{2\lfloor \frac{B}{R} \rfloor}{R^2 \epsilon} < \frac{2n^2 \lceil \log_2(n) \rceil^3}{B^2 \epsilon}
$$

by diving the first term by the second:

$$
\frac{2\lfloor \frac{B}{R} \rfloor}{R\epsilon} \Big/ \frac{2n\lceil \log_2(n) \rceil^2}{B\epsilon} = \lfloor \frac{B}{R} \rfloor \frac{B}{R} \cdot \frac{1}{n\lceil \log_2(n) \rceil^2} < 1.
$$

The last inequality holds because both $\frac{B}{R}$ and $\lceil \log_2(n) \rceil$ are the number of rounds and are considered the same.

## D  Computational Details

**Approximating $\Delta\sigma$ for Candidates.** In our probabilistic model, $\ell(\mathbf{y}, M_{\gamma_c}^*(\mathbf{X}))$ is approximated by a Gaussian distribution parameterized by $\mu_c$ and $\sigma_c$. To compute the uncertainty reduction that would result from training each candidate for one additional epoch, we consider the uncertainty as a time series in the form of $(\sigma_c(t))_{t=1}^T$. We examine the uncertainty trendings in NAS-BENCH-201 [9]. Figure 4 shows a certain pattern of uncertainty behavior as $t$ increases, both individually and aggregately, for different candidates.

We then model $(\sigma_c(t))_{t=1}^T$ according to different phases in the following way:

1. For $t \in (0, 6)$, $\sigma(t)$ increases. We use linear regression to fit the $\sigma(t)$, namely, $\sigma(t) = a_1 t + b_1$.

2. For $t \in (6, 50)$, $\sigma(t)$ drops quickly. We use the exponential model to fit $\sigma(t)$, namely, $\sigma(t) = a_1 e^{-b_1 x}$.

3. For $t \in (50, 180)$, $\sigma(t)$ increases steadily. We use another linear regression to fit $\sigma(t)$.

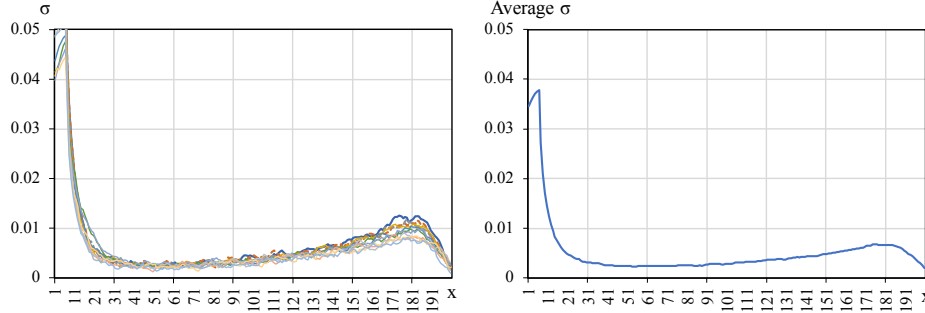

(a) Uncertainty scope for different candidates.   (b) Average uncertainty for different candidates.

Figure 4: Landscape of the uncertainty scope for different epochs.

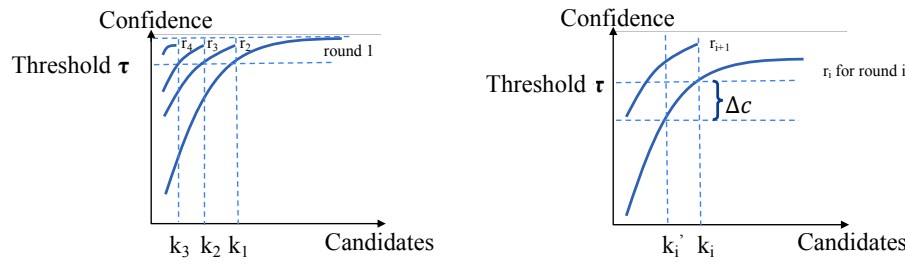

(a) Confidence curves in different rounds.   (b) Analysis of the choice of $\tau$.

Figure 5: Illustration for *confidence curve* and *discarding mechanisms*. After obtaining the *confidence curve*, a threshold $\tau$ determines the number of candidates we will keep ($k_1$ for round 1, $k_2$ for round 2, and $k_3$ for round 3). We choose the smallest $k$ such that $P_k \geq \tau$ for each round, proceed training with the best performed $k$ candidates, and discard the rest configurations.

    4. For $t \in (180, 200)$, $\sigma(t)$ reverses the trend and decreases again. We use linear regression.

For each launch, we sample a few candidates and train each one fully till convergence. We then use the abovementioned way to model the uncertainty behavior for the whole dataset. This makes the approximation for $\Delta_t \sigma = \sigma(t) - \sigma(t+1)$ effective and efficient.

An alternative way to approximate $\Delta_t \sigma$ is to use

$$\frac{1}{\delta} \sum_{b \in D_{t-1}} (\ell(\mathbf{y}, M_{\gamma_c}^b(\mathbf{X})) - \mathbb{E}_{D_{t-1}}[\ell(\mathbf{y}, M_{\gamma_c}^b(\mathbf{X}))])^2 - \frac{1}{\delta} \sum_{b \in D_t} (\ell(\mathbf{y}, M_{\gamma_c}^b(\mathbf{X})) - \mathbb{E}_{D_t}[\ell(\mathbf{y}, M_{\gamma_c}^b(\mathbf{X}))])^2.$$

**Building Probabilistic Model.** At any given time $t$, the approximation of converged validation loss follows the Gaussian distribution: $\ell(\mathbf{y}, M_{\gamma_c}^*(\mathbf{X})) \sim \mathcal{N}(\mu_c, \sigma_c^2)$. In our experiments, $\delta = 10$, $\mu_c$ is the current accuracy $\ell(\mathbf{y}, M_{\gamma_c}^t(\mathbf{X}))$, and $\sigma_c$ is the unbiased estimation of the standard deviation of $\ell(\mathbf{y}, M_{\gamma_c}^i(\mathbf{X}))_{i=t-10}^t$.

# E   Additional Related Work

**Freeze-thaw Bayesian optimization.** This paper proposes a new method to improve Bayesian optimization (BO) for HPO. It avoids the limits of the expected improvement (EI) criterion in naive BO which always favors picking new candidates rather than running old ones for more iterations. Instead of always sampling new candidates, it can also choose old candidates for further evaluation based on the modified EI in each round. This method, however, is limited to BO method. In contrast, our UQ scheme is applicable to the vast number of early stopped and multi-fidelity-based HPO methods.

**HyperJump** improves HB in that, during HPO, it skips certain rounds for certain candidates if the risk of skipping is within a threshold. For the NAS-BENCH-201 trained on ImageNet-16-120, HJ reduces the running time compared to the original HB method (only needs a 5.3% fraction of budget to achieve close to optimal results achieved by the original HB with a standard full budget), but it does not improve the HPO performance (i.e., Top-1 Rank and Regret). In contrast, our method only needs less than a 3% fraction of the budget to achieve close to optimal results and when using around 5% fraction of budget, our method reduces the regret by 33%.

# F   Benchmark and Dataset Information

Table 2 consolidates information on the datasets, hyperparameters, fidelity, and dataset sizes for Nas-Bench-201 and LCBench. The datasets for LCBench are drawn from various sources [37, 13].

Table 2: Benchmark and Dataset information.

| Tasks | Datasets | Hyperparameters | Fidelity | # Training set | # Validation set | # Test set |
|---|---|---|---|---|---|---|
| Nas-Bench-201 | CIFAR-10 | $1 \leftarrow 0$ | 1-200 | 25K images | 25K images | 10K images |
| | CIFAR-100 | $2 \leftarrow \{0, 1\}^*$ | | 50K images | 5K images | 5K images |
| | ImageNet-16-120 | $3 \leftarrow \{0, 1, 2\}^*$ | | 151.7K images | 3K images | 3K images |
| | | Range: {none, skip_connect, nor_conv_1x1, nor_conv_3x3, avg_pool_3x3} | | | | |
| LCBench | Fashion-MNIST | Batch size: [16, 512], log-scale | 1-50 | "Whenever possible, we use the given test split with a 33% test split and additionally use fixed 33% of the training data as validation split. In case there is no such OpenML task with a 33% split available for a dataset, we create a 33% test split and fix it across the configurations." [42] | | |
| | adult | Learning rate: $[1e^{-4}, 1e^{-1}]$, log-scale | | | | |
| | higgs | Momentum: [0.1, 0.99] | | | | |
| | jasmine | Weight decay: $[1e^{-5}, 1e^{-1}]$ | | | | |
| | vehicle | Number of layers: [1, 5] | | | | |
| | volkert | Maximum number of units per layer: [64, 1024], log-scale | | | | |
| | | Dropout: [0.0, 1.0] | | | | |

# G   More Results on Experiments

We include more results on NAS-Bench-201 and LCBench. For example, Figure 10 and 11 show the results on LCBench, where we proved the consistently better performance of the UQ-guided approaches than the UQ-oblivious methods on Fashion-MNIST. Figure 10 shows the results of the validation loss while Figure 11 demonstrates the results of regret. UQ-guided approaches obtained an average of over 50% improvement over the UQ-oblivious counterparts.

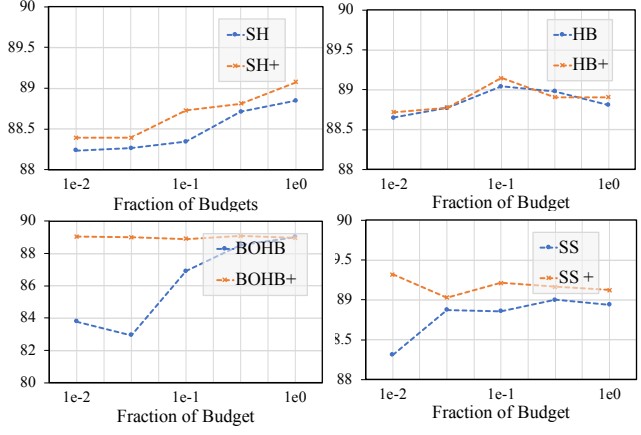

Figure 6: Results of test accuracy when optimizing on CIFAR-10.

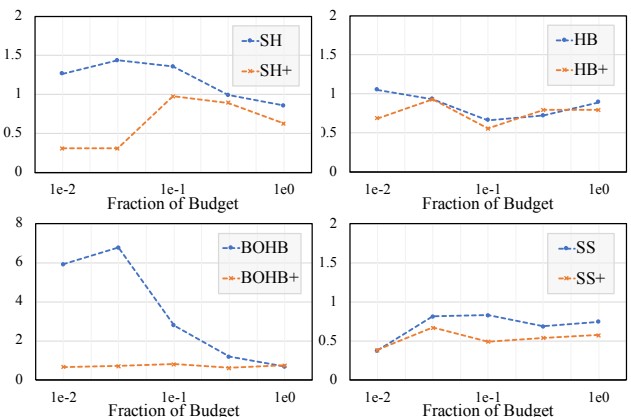

Figure 7: Results of regret (%) when optimizing on CIFAR-10.

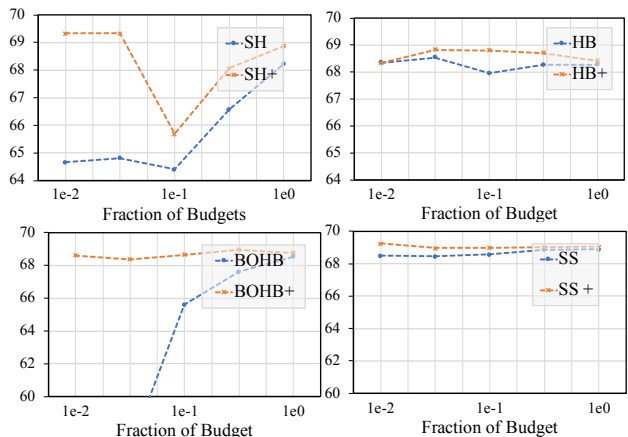

Figure 8: Results of test accuracy when optimizing on CIFAR-100.

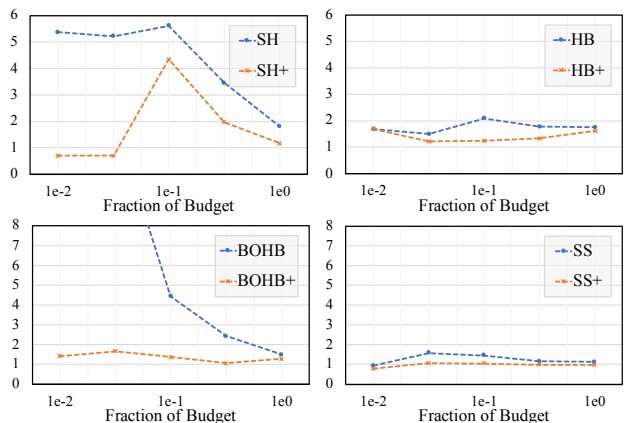

Figure 9: Results of test accuracy when optimizing on CIFAR-100.

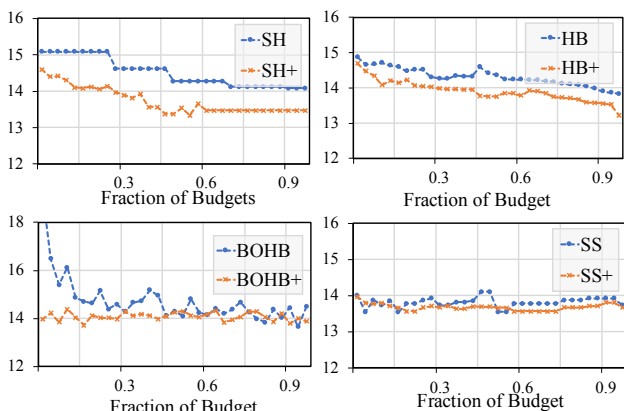

Figure 10: Results of validation error for optimizing on Fashion-MNIST.

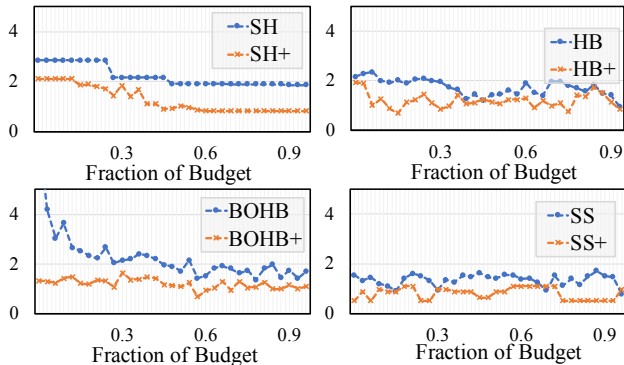

Figure 11: Results of regret (%) on test accuracy for optimizing on Fashion-MNIST.

# H  Experiments on Other Iterative Learners

We focus on DNNs in our experiments because (1) DNNs are among the most influential models today and (2) DNN training takes a long time so selecting the optimal hyperparameters is a critical concern, making the problem more pressing.

Even though we focus on DNN methods, our approach can be applied to other iterative learners. We consider a ridge regression problem trained with stochastic gradient descent on this objective function with step size $.01/\sqrt{2+T}$. The $l_2$ penalty hyperparameter $\lambda \in [10^{-6}, 10^0]$ was chosen uniformly at random on a log scale per trial. We use the Million Song Dataset year prediction task [27] with the same experiment settings as in the original SH paper. We show the results of ridge regression on "SH" and "SH+". Figure 12 shows the Top-1 Rank results and the regret of the test error for different fractions of budgets. The average results of 30 repetitions are reported. The benefits are obvious: SH+ obtained an average of over 40% improvement over the SH.

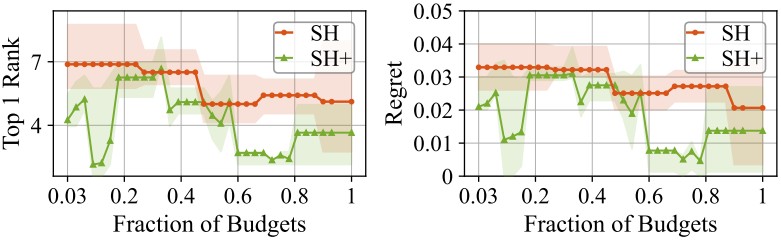

Figure 12: Results of Top-1 Rank and Regret on test error for optimizing ridge regression.

