# OpenReview forum: "UQ-Guided Hyperparameter Optimization for Iterative Learners"
_NeurIPS.cc/2024/Conference — NeurIPS 2024 poster_

### Official Review · Reviewer_ffRZ · 2024-07-02

**Soundness:** 3
**Presentation:** 3
**Contribution:** 3
**Rating:** 7
**Confidence:** 5

**Summary:**

The authors presented a novel scheme for hyperparameter optimization (HPO) in machine learning (ML) based on quantization uncertainty. The paper reflects a clear consequent way of discussion from introduction and literature overview to conclusions and very detailed appendixes. The authors demonstrated a strong mathematical background of these results with a full understanding uncertainly in machine learning model optimization process. Work presents theoretical analysis, examples and experiments on different related algorithms. Presented uncertainly quantization guided scheme can be widely applied for HPO in ML models optimization.

**Strengths:**

•	Universalizing provided improving technique (originality and quality).
•	Using different conventional HPO algorithms for demonstration novel approach (clarity and quality).
•	Strong mathematical background and proofs (quality, originality and significance).
•	Theorical analysis presented guided scheme (quality and significance).
•	Clear experiments methodology and results (clarity).

**Weaknesses:**

Perhaps authors should provide more detailed discussion about UQ-scheme limitations and adaptation for non-iterative learners and add some links about such learners

Mistypes and etc.
Appendix C, page 21, lines 2 and 3 in the first paragraph. Broken links.
Appendix D, page 24, line four in the first paragraph. Broken equation link.

**Questions:**

Has the UQ-scheme some limitations depending on the number of initial candidates for SH algorithm, for example? Could this be the case where UQ-scheme performance will be almost the same as in the original HPO algorithm, where regret between SH will equal SH+ with large or small vise-versa amount of the initial candidates? Or accuracy derivation for confidence curve is absolutely stable for any quantity of candidates?

**Limitations:**

Limitations this work mainly addressed to HPO in machine learning applications. Authors build adequate descriptions and analysis of their technique only within ML applications (abstract introduction and conclusion). Moreover, the authors noted that their HPO tuning scheme can be mostly suitable for iterative learners.

---

> ### Author Rebuttal · Authors · 2024-08-07
>
> >**Comment 1: Has the UQ-scheme some limitations depending on the number of initial candidates for SH algorithm, for example? Could this be the case where UQ-scheme performance will be almost the same as in the original HPO algorithm, where regret between SH will equal SH+ with large or small vise-versa amount of the initial candidates? Or accuracy derivation for confidence curve is absolutely stable for any quantity of candidates?**
>
> Response: We compare different numbers of initial candidates in the same setting for NATS-BENCH-201 on ImageNet.
>
> |                    |     SH     |        |     SH+    |        |
> |:------------------:|:----------:|:------:|:----------:|:------:|
> | Initial Candidates | Top-1 Rank | Regret | Top-1 Rank | Regret |
> |         50         |     4.2    |  5.06  |    1.96    |  2.52  |
> |         500        |    22.95   |   5.3  |    5.29    |  2.74  |
>
> When the total number of initial candidates increases, both SH and SH have a higher Top-1 Rank; the regret, however, stays stable, showing that the UQ-scheme SH+ scales well as the number of initial candidates increases.
>
> > **Comment 2: Perhaps authors should provide more detailed discussion about UQ-scheme limitations and adaptation for non-iterative learners and add some links about such learners.**
>
> Response: We will add a detailed discussion on the limitations of our proposed methods in the revision. Please see the global comment ``Limitations of the current work".

---

> > ### Comment · Reviewer_ffRZ · 2024-08-12
> >
> > Thanks for your responses. I will remain the score.

---

### Official Review · Reviewer_gRaV · 2024-07-09

**Soundness:** 3
**Presentation:** 2
**Contribution:** 2
**Rating:** 6
**Confidence:** 3

**Summary:**

The paper presents UQ-guided scheme, an approach for hyper-parameter optimisation (HPO) that quantifies uncertainty of each candidate configuration. This uncertainty quantification mechanism is applied to several state-of-the-art iterative HPO algorithms to prevent that promising configurations that have low performance in the early stages of the iterative process are not incorrectly discarded. The evaluation shows that UQ allows to find similarly performing configurations at a fraction of the exploration budget.

**Strengths:**

**Originality:**

The authors claim the method is new and that they only found one work that accounts for uncertainty (reference 30), however there are more works that account for uncertainty when performing hyper-parameter optimization. For example, the following:

[a] Mendes, Pedro, et al. "HyperJump: accelerating HyperBand via risk modelling." Proceedings of the AAAI Conference on Artificial Intelligence. Vol. 37. No. 8. 2023.

This other work may also be of relevance for the authors:

[b] Swersky, Kevin, Jasper Snoek, and Ryan Prescott Adams. "Freeze-thaw Bayesian optimization." arXiv preprint arXiv:1406.3896 (2014).

I would have liked to see a comparison of the proposed method against reference [30], given that it was the only work the authors found that accounted for uncertainty, and also against [a].

&NewLine;
**Quality:**

The claims seem well supported and the theoretical methodology sound. The methods used are appropriate. Weaknesses could be discussed in more detail and comparison agains related work could be improved/extended.

&NewLine;
**Clarity:**

Overall, the paper is well written. There are a couple of sentences here and there that could be improved, but it doesn't substantially affect understandability.

I believe the main manuscript should have pseudo-code illustrating the extra steps that should be added to any iterative learner that a user would like to change to account for model uncertainty. There is one example in the supplemental material but it seems to be specific to SH. What is generalisable from that example and what is not?

The manuscript is quite heavy on math and it seems like some parts could profit from an intuition-guided explanation. For example, in section 3.2, paragraphs "Decomposition of momentum and the underlying structure of the metric" and "Solving for the Momentum Mean and Variance" could have a more high-level description of why it is important to analyse those decomposed parts, what type on information is expected to be extracted from each, and how that information can contribute to improving the selection of the (near-)optimal configuration. Some math details can be relegated to the supplemental material, should space be a concern.

Finally, which parameters does the UQ approach have and that a user must set? How do those parameters affect the improvements attainable by UQ?

&NewLine;
**Significance:**

Improving the efficiency of hyper-parameter optimization is a relevant problem and efficient methods that are shown to substantially improve over existing methods are relevant. However, I felt the current version of the manuscript misses comparisons against recent related work.

**Weaknesses:**

See above for strengths and weaknesses.

**Questions:**

**a)**
Why do you claim that "data uncertainty is constant"? (This is in the beginning of section 3.1)

**b)**
Section 3.3. -- exploration vs exploitation: is k_1 - 1 == k'_i ?

**c)**
Section 3.4, Example 3 is not clear to me. In both cases you mention in the example, the UQ method "can return the best candidate with probability over 1 − n\delta". Do you mean that the probability of UQ returning the best candidate is larger than "1 − n\delta" (which is the probability of the UQ-oblivious method returning the best candidate)? Also, there is a 'But' in the middle of the example, but I don't understand the 'But' and I don't see where the expression  for B_{UQ} comes from.

**d)**
Which parameters does the UQ approach have and that a user must set? How do those parameters affect the improvements attainable by UQ?

&NewLine;
**Comments:**

- Section C.1 of the appendix has some missing pointers to assumptions.

- The evaluation plots would all be much easier to read if the y-axis was labelled.

- You could also highlight the moment when the UQ-version of each baseline achieves comparable performance to the UQ-oblivious version. This would make it easier to visually identify the gains achievable via UQ in terms of budget savings.

**Limitations:**

There could be more discussion on the limitations of the proposed method. For example, when is it applicable, how do the different assumptions impact its applicability, how could future work improve the current version of the proposal to address these limitations.

---

> ### Author Rebuttal · Authors · 2024-08-07
>
> >**Comment 1: Why do you claim that "data uncertainty is constant"? (This is in the beginning of section 3.1)**
>
> Response: In our setting, each candidate is trained on the same set of training data so the data uncertainty among different candidates is the same.
>
> > **Comment 2: Section 3.3. -- exploration vs exploitation: is $k_1 - 1 == k'_i$ ?**
>
> Response: $k'_i$ is just another name for $k_i -1$. So yes, $k_i -1 == k'_i$.
>
> > **Comment 3: Section 3.4, Example 3 is not clear to me. In both cases you mention in the example, the UQ method "can return the best candidate with probability over $1 - n\delta$". Do you mean that the probability of UQ returning the best candidate is larger than "$1 - n\delta$" (which is the probability of the UQ-oblivious method returning the best candidate)? Also, there is a 'But' in the middle of the example, but I don't understand the 'But' and I don't see where the expression for $B_{UQ}$ comes from.**
>
> Response: Yes, we mean the UQ method can return the best candidate with probability greater than $1-n\delta$.
> To guarantee the same probability of identifying the optimal candidate, the UQ-oblivious method needs a budget $B_{ob}$ greater than what the UQ method needs ($B_{UQ}$). `But' here stresses that $B_{UQ}$ is smaller than $B_{ob}$. The expression denotes a lower bound budget, specifically, $\sqrt[4]{2} \cdot \gamma^{-1}(\tfrac{\nu_2-\nu_1}{2}, \delta) \cdot n\simeq \gamma^{-1}(\tfrac{\nu_2-\nu_1}{2}, \delta) \cdot n$, for the UQ method to guarantee the 1-$n\delta$ success probability. This lower bound is proved in Corollary 6.
>
> > **Comment 4: Which parameters does the UQ approach have and that a user must set? How do those parameters affect the improvements attainable by UQ?**
>
> Response:
> Parameters for the UQ approach include the total budget and the predefined budget resources (e.g., training epochs) for each round. We set these parameters according to the same default value as in the paper HB.
> To quantify the uncertainty in the HPO process, we leverage the validation history to construct an estimated loss curve. This estimation uses parameters $\mathbf{k}_t$ to model the loss curve from $F_t$ samples for each candidate. More complex parameters for modeling the loss curve may better fit the current observed loss curve but at a risk of losing generality. $F_i$ is always small in our experiments because each sample requires training an entire model.
>
> >**Comment 5: There could be more discussion on the limitations of the proposed method. For example, when is it applicable, how do the different assumptions impact its applicability, how could future work improve the current version of the proposal to address these limitations.**
>
> Response: We will add a detailed discussion on the limitations of our proposed methods in the revision. Please see the global comment ``Limitations of the current work".
>
> >**Comment 6: The work may also be of relevance for the authors:
> [b] Swersky, Kevin, Jasper Snoek, and Ryan Prescott Adams. "Freeze-thaw Bayesian optimization." arXiv preprint arXiv:1406.3896 (2014).**
>
> Response: This paper proposes a new method to improve Bayesian optimization (BO) for HPO. It avoids the limits of the expected improvement (EI) criterion in naive BO which always favors picking new candidates rather than running old ones for more iterations.
> Instead of always sampling new candidates, it can also choose old candidates for further evaluation based on the modified EI in each round.
> This method, however, is limited to BO method.
> In contrast, our UQ scheme is applicable to the vast number of early stopped and multi-fidelity-based HPO methods.
>
> >**Comment 7: I would have liked to see a comparison of the proposed method against reference [30], given that it was the only work the authors found that accounted for uncertainty, and also against [a].
> [a] Mendes, Pedro, et al. "HyperJump: accelerating HyperBand via risk modeling." Proceedings of the AAAI Conference on Artificial Intelligence. Vol. 37. No. 8. 2023.**
>
> Response: Comparison with the HyperJump (HJ) work.
>
> HyperJump improves HB in that, during HPO, it skips certain rounds for certain candidates if the risk of skipping is within a threshold.
> For the NAS-BENCH-201 trained on ImageNet-16-120, HJ reduces the running time compared to the original HB method (only needs a 5.3\% fraction of budget to achieve close to optimal results achieved by the original HB with a standard full budget), but it does not improve the HPO performance (i.e., Top-1 Rank and Regret).
> In contrast, our method only needs less than a 3\% fraction of the budget to achieve close to optimal results and when using around 5\% fraction of budget, our method reduces the regret by 33\%.
>
> Comparison with [30].
> The performance gaps (regret) of [30] are lower than that of SH (a reduction of 30\%) but the execution time used for [30] is two times that of SH. This is because [30] needs to evaluate most of the models despite that it can skip some intermediate evaluations based on the upper-bound predictions.
> Our UQ method, however, can reduce 50\% of regret with the same budget resources as SH.
>
> >**Other comments on Clarity.**
>
> We will address these according to the comments and add a more high-level description for paragraphs "Decomposition of momentum and the underlying structure of the metric" and "Solving for the Momentum Mean and Variance".
>
> We will also include the pseudo-code for the other UQ-guided methods.
>
> The two assumptions in the proof of Theorem 1 are (1): $\ell(\mathbf{y}, M_{\gamma_c}^{*}(\mathbf{X})) = \lim\limits_{t\to \infty}\ell(\mathbf{y}, M_{\gamma_c}^{t}(\mathbf{X}))$  exists for $\gamma_c \in \Gamma$ and (2) $ \nu_i=\lim\limits_{\tau \to \infty} \ell_{i, \tau}$. These assumptions imply that the machine learning model will eventually converge after enough epochs. We will fix this in the revision.

---

> > ### Comment · Reviewer_gRaV · 2024-08-11
> >
> > Thank you for the clarifications.

---

### Official Review · Reviewer_P7Zr · 2024-07-12

**Soundness:** 2
**Presentation:** 3
**Contribution:** 3
**Rating:** 6
**Confidence:** 4

**Summary:**

In the paper "UQ-Guided Hyperparameter Optimization for Iterative Learners" the authors present an uncertainty quantification method for optimizing the hyperparameters of iterative learners in a multi-fidelity setting. It is argued that the best possible candidate is often mistakenly discarded at some point and uncertainty quantification can help avoid such mistakes. Furthermore, it can help to identify the best-performing candidate with certainty earlier than running the entire promotion scheme of, e.g., successive halving.

**Strengths:**

- The paper presents a novel and interesting approach to HPO considering uncertainties throughout the HPO process to incorporate them and making decisions more cautiously.
- The approach is theoretically grounded and the authors also provide a theoretical framework for their method.
- In experiments, the method also performs favorably compared to methods that do not leverage the information about uncertainty.

**Weaknesses:**

- The authors claim that their method is working for any kind of iterative learners. However, this claim is not supported in the experimental section as only deep learning methods are considered. Therefore, it is questionable whether the same quality of results could be observed for other iterative learners, e.g., logistic regression.
- Moreover, to make their approach work, the authors need a learner that is able to quantify its epistemic uncertainty, i.e., model uncertainty. However, I do not see how you would make this work for iterative learners in general where you cannot make use of simple tricks as in deep learning models to obtain ensembles through MCMC dropout or something similar.
- It could be made clearer that the authors specifically target multi-fidelity/early stopping methods with their method.
- The writing of the paper could be improved. For example, in the introduction the authors write about "steps" but it is not clear what is meant by steps, in particular, what are "steps of exploration". In general, there are some linguistic issues throughout the paper which the authors could easily get rid off using tools like Grammarly.
- In Section 3.2 the authors formalize their setting by attributing hyperparameters to a model, however, some hyperparameters belong to the learning algorithm, e.g., weight decay or learning rate, but not to the model.
- It is somewhat strange that the plus variants (including UQ) always start better than the basic method. In particular, it seems awkward for zero budget that the proposed method can better identify the top 1 rank than the base method. Also for the regret it is unclear how UQ can help identifying a more suitable candidate if nothing is known about this candidate.

**Questions:**

- How can the epistemic uncertainty be quantified for other learners?
- Why does the performance of X+ always improve already over X if there is no budget used?
- How is the uncertainty or let us say better certainty about the top 1 rank leveraged in the HPO tool? It would be quite straightforward to stop evaluating other candidates if there is enough evidence already that the top 1 candidate is correctly identified.
- If in every iteration of successive halving the amount of budget spent for this iteration is kept at a certain level, how is then the budget distributed if more candidates are to be evaluated in that iteration or could it also happen that more budget is assigned to every remaining candidate than originally planned according to e.g. successive halving?

**Limitations:**

Limitations are not sufficiently addressed. Although the authors mention in the questionaire that limitations would be addressed there is just a single sentence that it is mostly suitable for iterative learners and needs adaptions for other learners. This is definitely not a sufficient limitations discussion.

---

> ### Author Rebuttal · Authors · 2024-08-07
>
> > **Comment 1: The authors claim that their method is working for any kind of iterative learners. However, this claim is not supported in the experimental section as only deep learning methods are considered. Therefore, it is questionable whether the same quality of results could be observed for other iterative learners, e.g., logistic regression.**
>
> Response: We focus on DNNs in our experiments because (1) DNNs are among the most influential models today and (2) DNN training takes a long time so selecting the optimal hyperparameters is a critical concern, making the problem more pressing.
>
> Even though we focus on DNN methods, our approach can be applied to other iterative learners.
> We consider a ridge regression problem trained with stochastic gradient descent on this objective function with step size $.01/\sqrt{2 + T}$.
> The $l_2$ penalty hyperparameter $\lambda \in [10^{-6}, 10^0]$ was chosen uniformly at random on a log scale per trial.
> We use the Million Song Dataset year prediction task [1] with the same experiment settings as in the original SH paper.
> In the limited time, we managed to get the results of ridge regression on "SH" and "SH+".
> Figure 12 in the attached pdf in the global rebuttal shows the Top-1 Rank results and the regret of the test error for different fractions of budgets.
> The average results of 30 repetitions are reported.
> The benefits are obvious: SH+ obtained
> an average of over 40\% improvement over the SH.
>
> [1] Lichman, M. UCI machine learning repository, 2013. URL http://archive.ics.uci.edu/ml.
>
> >**Comment 2: Moreover, to make their approach work, the authors need a learner that is able to quantify its epistemic uncertainty, i.e., model uncertainty. However, I do not see how you would make this work for iterative learners in general where you cannot make use of simple tricks as in deep learning models to obtain ensembles through MCMC dropout or something similar.
> How can the epistemic uncertainty be quantified for other learners?**
>
> Response: For other iterative learners (e.g., regression problem trained with stochastic gradient descent on the objective function with step size and $\ell_2$ penalty hyperparameter $\lambda$), we can leverage the loss curves of the training process of the models we have already observed and use the formula in page 4 to estimate the epistemic uncertainty.
>
> >**Comment 3: Why does the performance of X+ always improve already over X if there is no budget used?**
>
> Response: We have updated Figure 3 with the fixed $x$-axis label in the attached pdf in the global rebuttal. There was a minor typo in the figure: The label 0 in the $x$-axis was supposed to be 0.03.
> Figure 3 shows that the UQ-guided approach can always achieve better Top-1 Rank and regret results than the UQ-oblivious method does even with a small budget (3\% of the standard full budget).
>
> >**Comment 4: How is the uncertainty or let us say better certainty about the top 1 rank leveraged in the HPO tool? It would be quite straightforward to stop evaluating other candidates if there is enough evidence already that the top 1 candidate is correctly identified.**
>
> Response: The key issue is that identifying the top-1 candidate during the HPO process is challenging.
> As shown in Figure 1, candidates that initially appear to be top-1 often turn out to be inferior after convergence. Our entire paper, therefore, focuses on how to quantify this uncertainty of the model quality during the intermediate steps of HPO and integrate this uncertainty into the selection process to maximize the likelihood of choosing the right candidate.
>
> >**Comment 5: If in every iteration of successive halving the amount of budget spent for this iteration is kept at a certain level, how is then the budget distributed if more candidates are to be evaluated in that iteration or could it also happen that more budget is assigned to every remaining candidate than originally planned according to e.g. successive halving?**
>
> Response: As mentioned in Figure 2, the budget is evenly assigned among candidates in each round, so each candidate will adopt $\frac{R}{K}$ budget resource ($R$ is the round budget and $K$ is the number of candidates kept in this round). That is to say, if, according to our probabilistic model, we need to keep more candidates to evaluate, then the budget for each candidate in this round is reduced. Likewise, more budgets are assigned to each one if fewer candidates remain.
> Note that we determine the number of candidates to keep by considering the tradeoff between the risks of discarding the best candidate and the training budget each top candidate can get.
>
> >**Comment 6: In the introduction the authors write about "steps" but it is not clear what is meant by steps, in particular, what are "steps of exploration".**
>
> Response: "8-22 steps of the exploration" here indicate that the best candidate is discarded after that number of iterations of training (because its validation loss is not in the better half among all the candidates that still remain).
> To make it clear and consistent with Figure 1, we will change the term "steps of exploration" to "iterations of training". We will also fix other linguistic issues in the paper.
>
> >**Comment 7: In Section 3.2 the authors formalize their setting by attributing hyperparameters to a model, however, some hyperparameters belong to the learning algorithm, e.g., weight decay or learning rate, but not to the model.**
>
> Response: We will replace "a model" with "a candidate" to denote a candidate hyperparameter configuration that can include hyperparameters for both the model and the learning algorithm.

---

> > ### Comment · Reviewer_P7Zr · 2024-08-12
> > **Response to Rebuttal**
> >
> > Thank you very much for the detailed response and the clarification.
> >
> > Ad Response to Comment 1:
> > Why is the step size fixed in this example? Would not it make sense to tune the step size too?
> >
> > Ad Response to Comment 2:
> > How would the loss curves be leveraged to extract the epistemic uncertainty? Learning curves can be pretty noisy and are also affected from the chosen hyperparameter values such as the step size. Just imagine a step size chosen by the tuner that is way too big. To me, it is not entirely clear how the aleatoric uncertainty is disentangled from the epistemic uncertainty and as the paper is claimed to work for iterative learners in general, for me, this is an essential part of the paper to be demonstrated. If the title of the paper was for deep learning methods, I would be satisfied with the scope of the paper.
> >
> > Ad Response to Comment 4:
> > Yes I got that from the paper. But if the quantification yields that the best candidate is identified with a probability of 99%, would not it make sense to spare resources and stop the evaluation?
> >
> > Ad Response to Comment 5:
> > Got it. But could not this flaw the overall HPO process? If you for example encounter a setting where you can never certainly drop any candidate, either due to very similar performance or large uncertainty bands, you will not return a model that is trained on the full assignable budget, do you?

---

> > > ### Author Response · Authors · 2024-08-14
> > > **Response to Reviewer P7Zr**
> > >
> > > >**Commet 8: (Response to Comment 1) Why is the step size fixed in this example? Would not it make sense to tune the step size too?**
> > >
> > > Response: We follow the original benchmark setting where we set the decreasing step size according to epochs and tune the penalty parameter.
> > > The step size can be tuned as well.
> > > By applying different step schedulers, we can further expand the search space.
> > >
> > > >**Comment 9: (Response to Comment 2) How would the loss curves be leveraged to extract the epistemic uncertainty? Learning curves can be pretty noisy and are also affected from the chosen hyperparameter values such as the step size. Just imagine a step size chosen by the tuner that is way too big. To me, it is not entirely clear how the aleatoric uncertainty is disentangled from the epistemic uncertainty and as the paper is claimed to work for iterative learners in general, for me, this is an essential part of the paper to be demonstrated. If the title of the paper was for deep learning methods, I would be satisfied with the scope of the paper.**
> > >
> > > Response: The “Solving for the Momentum Mean and Variance” part from the bottom of Page 3 to Page 4 explains how the uncertainty of the model’s predictions is estimated. Essentially, it strives to approximate the variations of the loss of the model at any given epoch, including the distribution of the loss of the converged model, and hence the uncertainty of its predictions. Details are described in that part of the paper.
> > >
> > >
> > >
> > > It is important to note that it is a statistical prediction process based on history. As with any typical statistical prediction, there is influence of some kinds of inseparable noises from various sources. The default schedule in our experiments already uses large learning rates at the beginning; fluctuations of loss curves at the early stage of HPO are commonly observed in our experiments. The existence of such influence is why empirical evaluations are necessary to validate whether the statistical prediction process could practically deliver good (of course, imperfect) predictions, which is what the results in our paper demonstrate.
> > >
> > >
> > >
> > > The experiments in our paper concentrate on DNN due to the importance of efficient HPO to the time-consuming nature of DNN training. Our ridge regression experiment in the rebuttal demonstrates the applicability of the methodology to other iterative learners. But because there was very limited time in the rebuttal period, the added experiment is a demonstration rather than a series of systematic experiments. We can understand the point of view of the reviewer about the scope of the paper. For the final version, we can narrow the main claims to DNN and use the ridge regression as a demonstration to show the potential of the methodology for other iterative learners, and leave a systematic study of which to the future.
> > >
> > > >**Comment 10: (Response to Comment 4) If the quantification yields that the best candidate is identified with a probability of 99\%, would not it make sense to spare resources and stop the evaluation?**
> > >
> > > Response: It absolutely makes sense, and our approach is designed to incorporate this principle.
> > > If the quantification indicates that the best candidate is identified with a sufficiently high probability (exceeding $\tau_i$ in round $i$), only this top candidate will be retained for that round. At this point, the HPO process is considered completed, and the remaining budget resources can be spared, as there is no need to evaluate additional candidates when only one remains.
> > >
> > > > **Comment 11: (Response to Comment 5) Could not this flaw the overall HPO process? If you for example encounter a setting where you can never certainly drop any candidate, either due to very similar performance or large uncertainty bands, you will not return a model that is trained on the full assignable budget, do you?**
> > >
> > > Response: If candidates have very similar performance or large uncertainty, our approach will retain
> > > $k$ candidates whose total probabilities of being the best candidate meet the threshold $\tau_i$, dropping the rest for that round. As rounds progress, this strategy exponentially reduces the number of candidates, efficiently narrowing down to the best options.

---

### Official Review · Reviewer_BPGo · 2024-07-14

**Soundness:** 3
**Presentation:** 3
**Contribution:** 3
**Rating:** 6
**Confidence:** 3

**Summary:**

This paper proposed an incorporate uncertainty quantification for hyperparamter tuning (learning rate, neural architecture). It is assumed the model performance metrics of interest, such as validation error, are Gaussian, and we can use the training output of first $N$ epochs to estimate the mean and variance of the performance metrics distribution. Using the distribution, this paper proposes a UQ-guided scheme by using confidence curves to find the highest performant combinations of hyperparameters.

**Strengths:**

It is shown by theory and empirical study that the proposed UQ method can enhance the existing hyperparameter tuning method like successive halving, which suffers from model uncertainty and incorrectly eliminating highly performant hyperparameter values

**Weaknesses:**

Estimating the mean and variance of the performance metrics curve may need to take some epochs.

**Questions:**

1. On page 4, it is stated that $\hat v$ and $\hat\sigma^2$ are estimated with N=200 epochs. My understanding is that the first N=200 epochs are only for estimating $\hat v$ and $\hat\sigma^2$ and the UQ-guided HPO are not used in these 200 epochs. However, I suspect we don’t have the luxury of running more than 200 epochs in some applications with large data.

2. The $Z$ in eq (3.1) seems to model a latent effect underlying the hyper parameter space. $Z$ is initialized randomly and then updated using the equation on p.4. No further analysis about $Z$ is provided in later sections, for example, whether we can use $Z$ to measure the contribution of certain combinations of hyperparameters. If estimating $Z$ explicitly doesn’t yield any useful products, perhaps there is no need to estimate Z and we just need to estimate $\alpha_u$?

3. Theorems:
- How to interpret the assumptions of Theorem 1?
- Does Theorem 2 apply to “any” UQ oblivious method like SH, HB? If so, it may be good to comment on/verify whether these existing methods satisfy the assumptions of Thm 2.

4. Can more details about $z_ob$ in Theorem 2 be given, e.g. how to arrive at the inequality? It is unclear how to interpret this quantity.
The proof of Theorem 1 refers to two assumptions but it looks the pdf was not compiled correctly so the reference hyperlinks are missing

**Limitations:**

The work is regarding a general algorithm aiming at optimizing HPO and belongs to a foundational research; its societal impacts are neutral.

---

> ### Author Rebuttal · Authors · 2024-08-07
>
> >**Comment 1: On page 4, it is stated that
> $\hat{v}$ and $\sigma^2$ are estimated with N=200 epochs. My understanding is that the first N=200 epochs are only for estimating $\hat{v}$ and $\sigma^2$ and the UQ-guided HPO are not used in these 200 epochs. However, I suspect we don’t have the luxury of running more than 200 epochs in some applications with large data.**
>
> Response: With our method, after any number of epochs, we can use the formula on page 4 to estimate the loss and uncertainty for any $N$ values. We do not need to wait for $N$ epochs before estimating. We plug in $N=200$ to the formula to estimate the uncertainty value at the $200$ epochs when most models would have already entered the convergence stage.
>
> >**Comment 2: The $Z$ in eq (3.1) seems to model a latent effect underlying the hyper parameter space. $Z$ is initialized randomly and then updated using the equation on p.4. No further analysis about $Z$ is provided in later sections, for example, whether we can use $Z$ to measure the contribution of certain combinations of hyperparameters. If estimating
> $Z$ explicitly doesn’t yield any useful products, perhaps there is no need to estimate $Z$ and we just need to estimate
> $\alpha^{\mathbf{u}}$?**
>
> Response: Estimating $Z$ is an indirect way to estimate $\alpha^{\mathbf{u}}$ in our setting so it allows correlation between the candidates.
> We can also estimate $\alpha^{\mathbf{u}}$ directly for each candidate; the experiments show satisfying results as well.
>
> > **Comment 3: Theorems:
> How to interpret the assumptions of Theorem 1?
> Does Theorem 2 apply to “any” UQ oblivious method like SH, HB? If so, it may be good to comment on/verify whether these existing methods satisfy the assumptions of Thm 2.**
>
> Response: Interpretation of the assumptions of Theorem 1:
>
> The assumption $ \nu_i=\lim\limits_{\tau \to \infty} \ell_{i, \tau}$ implies that the machine learning model will eventually converge after enough epochs. $\nu_i$ here denotes the converged loss for the machine learning model.
>
> Theorem 2 analyzes the UQ-oblivious early stop approaches and can be applied to methods such as HB as long as there are intermediate results available for comparison.
> We only make assumptions that the machine learning models will eventually converge after enough epochs and without loss of generality, these models are ordered according to their converged loss, namely, $\nu_1 \le \nu_2 \le \cdots \le \nu_n$.
> We make these assumptions regardless of the HPO method and they are easy to verify by looking into individual candidate's validation loss curve.
>
> >**Comment 4: Can more details about $z_{ob}$ in Theorem 2 be given, e.g. how to arrive at the inequality? It is unclear how to interpret this quantity. The proof of Theorem 1 refers to two assumptions but it looks the pdf was not compiled correctly so the reference hyperlinks are missing.**
>
> Response: The representation of $z_{ob}$ on the right-hand-side of the inequality is very intuitive:
> For each $i$, to merely verify that the $i$th candidate's final loss is higher than the best candidate's with a probability larger than or equal to $1-\delta$, one must train each of the two candidates at least a number of steps equal to the $i$th term in the sum.
> Repeating this argument for all $i$ explains the sum over all candidates.
>
>
> The two assumptions in the proof of Theorem 1 are (1): $\ell(\mathbf{y}, M_{\gamma_c}^{*}(\mathbf{X})) = \lim\limits_{t\to \infty}\ell(\mathbf{y}, M_{\gamma_c}^{t}(\mathbf{X}))$  exists for $\gamma_c \in \Gamma$ and (2) $ \nu_i=\lim\limits_{\tau \to \infty} \ell_{i, \tau}$. As addressed in the previous question, these assumptions imply that the machine learning model will eventually converge after enough epochs.
> We will fix that in the revision.

---

> > ### Comment · Reviewer_BPGo · 2024-08-11
> >
> > Thank you for answering my questions.

---

### Author Rebuttal · Authors · 2024-08-07

We thank the reviewers for the insightful feedback and suggestions. We address the common concern of the reviewers here:

## Limitations of the current work


The key characteristic of the UQ method is the necessity to rank multiple learners
during the HPO process. Gradient-based HPO methods [1], for instance, may not benefit
from our UQ-guided scheme because of their sequential properties.

### For non-iteration learners
The current version of our method is most suitable for iterative learners. To go
beyond, it could be, for instance, applied to the model selection
work in previous studies [2] that use training dataset size as the budget dimension. In
this case, the learner does not need to be iterative; the selection is based on the validation
loss history trained with incremental dataset sizes. The UQ component can still guide
the configuration selection and budget allocation in the HPO process.

[1] Micaelli, Paul, and Amos J. Storkey. ``Gradient-based Hyperparameter Optimization Over Long Horizons.” 34th Advances in Neural Information Processing Systems (NeurIPS). 2021.

[2] Mohr, Felix, and Jan N. van Rijn. ``Towards model selection using learning curve
cross-validation.” 8th ICML Workshop on automated machine learning (AutoML). 2021.

---

### Decision · Program_Chairs · 2024-09-25

**Decision:**

Accept (poster)

**Comment:**

This paper studies the uncertainty of the metric estimation in hyperparameter optimization. It introduces the concept of uncertainty-aware HPO and proposes the UQ-guided scheme for quantifying uncertainty. The proposed method applies to a board arrange of iterative learning methods and shows strong empirical performance.

All reviewers appreciate its theoretical analysis of the proposed method and the convincing empirical performance. Some reviewer also considers it a strength being versatile to apply to different conventional HPO algorithms.

There were some concerns in the original reviewers as follows,
1. Estimating the mean and variance of the performance metrics curve may need to take some epochs, which is not feasible with large datasets
2. The proposed method is limited to deep learning algorithms, in contrast to the claimed any iterative learner
3. Question about estimating the epistemic uncertainty decoupled from the aleatoric uncertainty
4. Lack of discussion and comparison with related work that also considers metric uncertainty

The authors feedback resolved many of those concerns, and two reviewers raised their rating to weak accept after the rebuttal, mainly due to its strong empirical performance in the experiments. The reviewers reach a consensus of acceptance.

Reviewers' concerns about point 1 and 2 above are not fully resolved from authors' response but are not considered major weaknesses against publication. I would encourage the authors to discuss the limitations more explicitly in the final revision. Also, please incorporate the clarification in the rebuttal, including the comparison with missed related work in the original version.

The following is additional comments from reviewers during the reviewer-AC discussion:

*"I still have doubts about the general applicability of the approach to any kind of iterative learner, defining a too broad scope by the title of the paper which has not been properly understood by the authors as I raised this point already in a previous revision (at another venue) of the paper.*"

"*I worry about the theoretical guarantees of approaches like SHA or Hyperband and whether they can maintain these guarantees with the changed promotion or budget allocation scheme."*

"*they didn't address the question that if $\hat{v}$ and $\hat{\sigma}^2$ and are estimated with just a few epochs, would one still observe the performance boosting. The concern I have is the performance may be unstable if and are estimated with only a few epochs.*"